# Dysfunctional β-cell longevity in diabetes relies on energy conservation and positive epistasis

Kavit Raval[1],*  , Neema Jamshidi[2],*  , Berfin Seyran[1]  , Lukasz Salwinski[3]  , Raju Pillai[4], Lixin Yang[4]  , Feiyang Ma[3], Matteo Pellegrini[3], Juliana Shin[5], Xia Yang[5], Slavica Tudzarova[1]

**Long-lived PFKFB3-expressing β-cells are dysfunctional partly because of prevailing glycolysis that compromises metabolic coupling of insulin secretion. Their accumulation in type 2 diabetes (T2D) appears to be related to the loss of apoptotic competency of cell fitness competition that maintains islet function by favoring constant selection of healthy "winner" cells. To investigate how PFKFB3 can disguise the competitive traits of dysfunctional "loser" β-cells, we analyzed the overlap between human β-cells with bona fide "loser signature" across diabetes pathologies using the HPAP scRNA-seq and spatial transcriptomics of PFKFB3-positive β-cells from nPOD T2D pancreata. The overlapping transcriptional profile of "loser" β-cells was represented by down-regulated ribosomal biosynthesis and genes encoding for mitochondrial respiration. PFKFB3-positive "loser" β-cells had the reduced expression of HLA class I and II genes. Gene–gene interaction analysis revealed that PFKFB3 rs1983890 can interact with the anti-apoptotic gene MAIP1 implicating positive epistasis as a mechanism for prolonged survival of "loser" β-cells in T2D. Inhibition of PFKFB3 resulted in the clearance of dysfunctional "loser" β-cells leading to restored glucose tolerance in the mouse model of T2D.**

## Introduction

Despite the clear link between accumulation of injured and dysfunctional β-cells and type 2 diabetes (T2D), it is difficult to target specifically these pathogenic cells in a therapeutic attempt to modify the trajectory of T2D. We reasoned that an effective approach would be to probe the existence of a context-dependent tissue property in health to which we can attribute the specific recognition and clearance of all injured and/or dysfunctional cells agnostic to their identity. As such, cell fitness competition (CFC) is emerging as a tissue clearance mechanism that maintains function by monitoring tissue behavior at the population scale (Merino et al, 2016) during and after development (Moreno et al, 2002; de la Cova et al, 2004; Gibson & Perrimon, 2005) and in adult post-mitotic tissues (Coelho & Moreno, 2019; Coelho et al, 2018; Vieira et al, 2024) (reviewed in Bowling et al [2019]). The process relies on the differential fitness within apparently isogenic cell population across several levels of cellular organization—from cellular resources (ribosomal biosynthesis, RiBi) to energy (metabolism) and infrastructure (mitochondria) (Coelho et al, 2018; Coelho & Moreno, 2019). Under homeostatic conditions, CFC enriches the tissue with functional and healthy cells without marked changes in tissue mass, which is deemed a silent phenotype (Blaauw et al, 2010; Sasai et al, 2010; Leychenko et al, 2011; Tamori & Deng, 2014). When damaged beyond repair, a cell's "molecular fitness fingerprint" is marked by protein aggregates and oxidative stress (Baumgartner et al, 2021). Mitochondrial dysfunction is common to different "loser" cells, and it is sufficient and necessary to trigger CFC (Lima et al, 2021). As such, "loser" epiblast cells undergo a transcriptional program in response to the impaired mitochondrial function that involves integrated stress response and unfolded protein response (UPR), implicating DNA Damage Induced Transcript 3 (Ddit3), Activating Transcription Factor 3 (Atf3), Protein Phosphatase 1 Regulatory Subunit 15A (Ppp1r15a) (Lima et al, 2021), and NFE2 Like BZIP Transcription Factor 2 (Nfe2l2) (Melber & Haynes, 2018; Munch, 2018; Rosario et al, 2020). Breaking down the "loser-to-winner" trajectory revealed that ribosomal synthesis and the targets of RPTOR Independent Companion Of MTOR Complex 2 (RICTOR) and MYC primarily fell within the down-regulated genes, constituting the "loser" fingerprints (Lima et al, 2021).

Our previous work has demonstrated that β-cells from T2D donors have a wide range of abnormalities that mirror those that are found in molecular "loser fingerprints," such as the proteotoxicity, UPR, and mitochondrial attenuation (Montemurro et al, 2019). Injured "loser" β-cells can escape removal by CFC (Bowling et al, 2019; Baker, 2020; Lawlor et al, 2020) by increasing the

[1]Hillblom Islet Research Center, David Geffen School of Medicine, University of California Los Angeles, Los Angeles, CA, USA   [2]Radiological Sciences, David Geffen School of Medicine, University of California Los Angeles, Los Angeles, CA, USA   [3]Molecular Cell and Developmental Biology, College of Life Sciences, University of California Los Angeles, Los Angeles, CA, USA   [4]Department of Pathology, City-of-Hope, Duarte, CA, USA   [5]Department of Molecular and Medical Pharmacology, University of California Los Angeles, Los Angeles, CA, USA

Correspondence: STudzarova@mednet.ucla.edu
*Kavit Raval and Neema Jamshidi contributed equally to this work

  

competitive advantage to "winner" cells via aerobic glycolysis using 6-phosphofructo-2-kinase/fructose-2,6-bisphosphatase 3 (PFKFB3) (Montemurro et al, 2019; Nomoto et al, 2020; Min et al, 2022). This sequence of observations implied that proteotoxicity, CFC, and functional cell regeneration could be linked, with T2D potentially representing a disease of dysfunctional cells that supposed to die but don't.

The link between PFKFB3 and CFC was indicated in the human-like murine model of T2D after β-cell depletion of PFKFB3 (*PFKFB3^BKO* on *diabetogenic stress, DS, proteotoxic stress [β-hTG], and high-fat diet*). PFKFB3 knockout led to an initial β-cell apoptotic competency and almost complete clearance of injured β-cells followed by near-wild-type levels of β-cell replication that resulted in maintained β-cell mass. The glucose tolerance restoring the phenotype of the *PFKFB3^BKO* DS mouse (under moderate DS stress) in connection to the elimination of injured ("loser") cells and silent β-cell mass phenotype closely reflected activated CFC, implicating PFKFB3 as a gatekeeper of CFC (Min et al, 2022).

We used dual criteria to investigate human "loser" β-cells based on (1) a bona fide "loser" signature adopted from the mouse embryo (Lima et al, 2021) and (2) PFKFB3 expression from our spatial transcriptomic analysis. We found no evidence for differential stress response between "loser" and "winner" β-cells. A notable exception was the general down-regulation of genes coding for ribosomal proteins and mitochondrial respiration, which eclipsed the impact of any other insufficiencies. The "long-lived" PFKFB3-positive β-cells demonstrated a complete transcriptional overlap with "loser" β-cells with extreme energy conservation and underlying aborted β-cell competition. From the available T1D and T2D single nucleotide polymorphism (SNP) pool of the HPAP database of Genotypes and Phenotypes (dbGAP), PFKFB3 polymorphism *rs1983890* interacted specifically with collective SNPs of ArfGAP with GTPase domain, ankyrin repeat and PH domain 1 (*AGAP1*), and the anti-apoptotic gene, the matrix AAA peptidase interacting protein 1 (*MAIP1*) (Opalinska & Janska, 2018).

We propose that the RiBi and mitochondrial respiration represent a "global" phenotypic interface of β-cell fitness. We also propose that PFKFB3-MAIP1 interaction can create an epiphenomenon of positive epistasis, which can be aborted by PFKFB3 inhibition. As such, inhibition of PFKFB3 led to the clearance of dysfunctional β-cells and restored glucose tolerance in the murine model of T2D, indicating that strategies to unlock islet CFC are viable alternatives for functional β-cell regeneration in T2D.

# Results

### Elucidating the "loser" β-cell trajectory in diabetes

Selective elimination of PFKFB3-positive β-cells by reactivation of CFC holds potential for islet enrichment by functional β-cell regeneration (Fig 1A). Therefore, we investigated a conceptual paradigm that diabetogenic stress counters the clearance of dysfunctional β-cells by a PFKFB3-dependent mechanism (Fig 1A). β-Cell dysfunction is linked to proteotoxicity and oxidative stress, interestingly, phenocopying the "loser" cell status established

previously (Baumgartner et al, 2021). We undertook a comprehensive genomic analysis of two independent datasets (67 donors from HPAP and 3 donors from nPOD) (Fig 1B). Datasets of differentially expressed genes (DEGs) were obtained by the module score–integrated R script based on bona fide "loser" genes: *DDIT3, ATF3, PPP1R15A, RICTOR,* and *NFE2L2* (Lima et al, 2021) across the disease spectrum from control (Control^HPAP), prediabetes (AAB^HPAP), type 1 diabetes (T1D^HPAP), to type 2 diabetes (T2D^HPAP) and including PFKFB3-expressing β-cells (T2D^PFKFB3) (genomic pipeline, Fig 1B). Interestingly, this combined approach yielded hundreds to thousands of DEGs in all disease states except T1D, where DEGs (adjusted $P < 0.05$) were represented by a skewed list and were not submitted to gene set enrichment analysis (GSEA). The T1D^HPAP DEG list comprised genes involved in immunity such as *CD81*, immunoproteasome component Proteasome 20S Subunit Beta 8 (*PSMB8*), cell structure regulators: Mitotic Spindle Organizing Protein 2B (*MZT2B*), a part of the gamma-tubulin complex, Translocase Of Outer Mitochondrial Membrane 6 (*TOMM6*), microsomal glutathione S-transferase 3 (*MGST3*), Ribonuclease K (*RNASEK*), TATA-Box Binding Protein Associated Factor 10 (*TAF10*), tetra-tricopeptide repeat protein 32 (*TTC32*), and others (Table S3). In addition, Control^HPAP did not yield significant enrichment in any pathway (Table S1). For the geospatial profiling, we have applied segmentation to document sequenced genes from 36–48 groups, each comprised 100 regions of interest (ROIs) of PFKFB3-positive and PFKFB3-negative β-cells (Fig S1A–F). Principal component analysis (PCA) predictably indicated proximity between the two β-cell subpopulations discriminated only by PFKFB3 expression (Fig S1G).

To find out whether PFKFB3 is a gatekeeper of CFC in diabetes (Montemurro et al, 2019; Nomoto et al, 2020), we sought to establish whether PFKFB3-positive β-cells are identical to "loser" β-cells from CFC. To conceptualize our objective, we considered β-cells with different competitive traits depicted in Fig 1C. "Loser" β-cells (L) that do not express PFKFB3 are elective for CFC. However, "loser" β-cells that express PFKFB3 (P+L) are non-elective by CFC because of PFKFB3 protection. Alone PFKFB3 (P) expression represents an indolent CFC status because it does not co-occur with the "loser" signature (Fig 1C).

GSEA based on DAVID and using Reactome as a reference unraveled differential transcriptomes ($P < 0.05$) of "loser" β-cells from Control^HPAP, AAB^HPAP, T1D^HPAP, T2D^HPAP, and T2D^PFKFB3 (Fig 1D and E). DEGs submitted to GSEA were normalized with the union of all genes from the HPAP database or the union of all genes from the nPOD database, respectively (Fig 1D and E and Tables S1, S2, S3, S4, and S5). The percentage coverage (the ratio of the observed gene to the background gene count) increased with unfolding of the T2D phenotype, whereas the strength of enriched pathways decreased from prediabetes (AAB^HPAP) to T2D^HPAP with T2D^PFKFB3 accounting for an intermediate state (Figs 1D and S2). The pairwise overlap between each of the two disease states is presented in Fig S3A–C.

Enriched pathways referred to the genes encoding for proteins involved in RiBi, protein translation, and peptide elongation processes in all datasets, whereas enriched pathways of mitochondrial respiration dominated the T2D^HPAP dataset (Fig S4). Among the RiBi-related enriched pathways in T2D^PFKFB3, AAB^HPAP, and T2D^HPAP, we found specifically *Protein translation, Peptide chain elongation,*

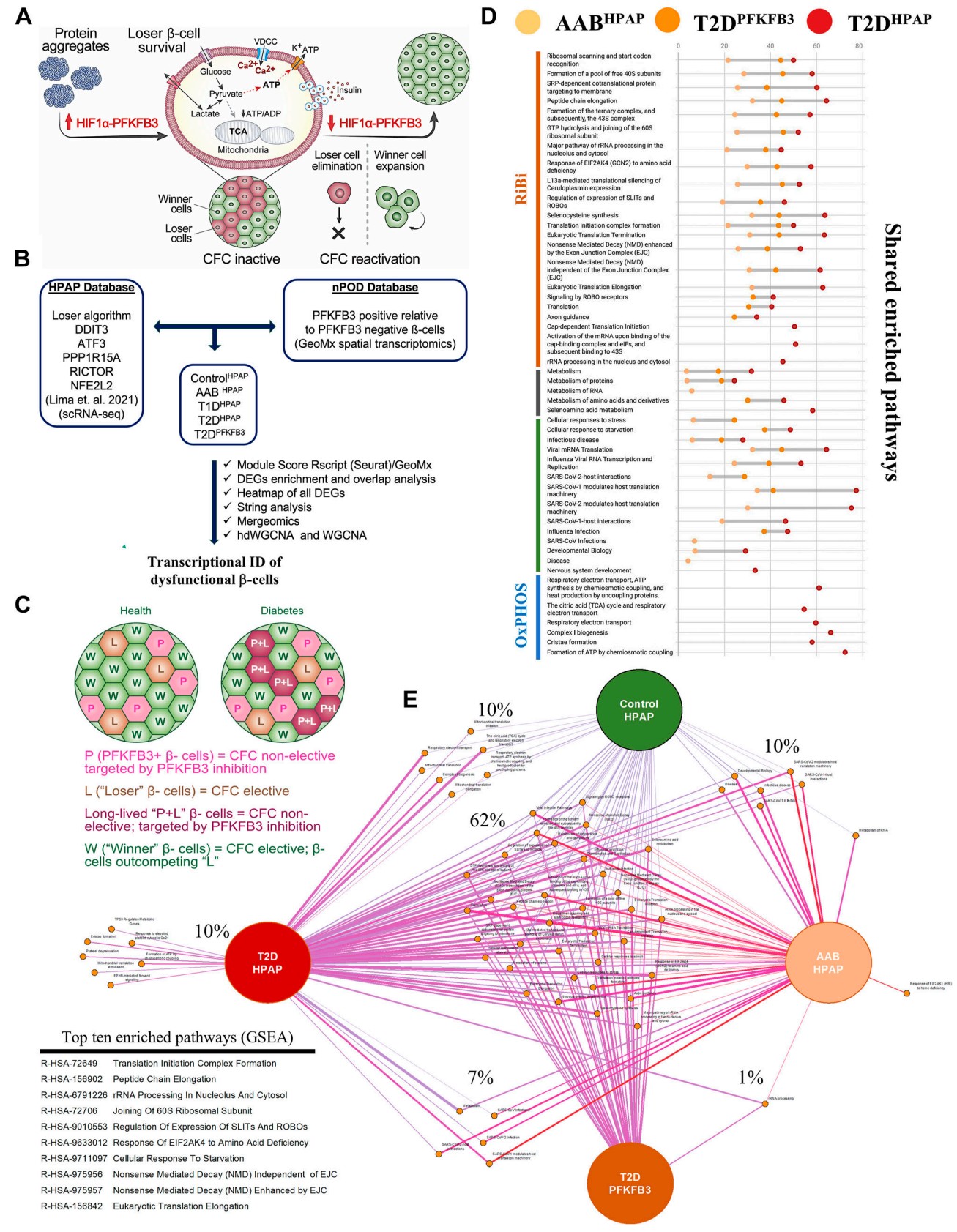

*Selenoamino acid metabolism, Signaling by ROBO receptors, Nonsense Mediated Decay (NMD)*, and many others, all presented in Fig S4 confirming alignment with the data in Fig 1D.

The DEGs from all datasets were dominated by down-regulated genes (Tables S1, S2, S3, S4, and S5). Before filtering by adjusted *P*-value, individual DEGs in AAB[HPAP], T2D[PFKFB3], and T2D[HPAP] showed global overlap in the direction of gene expression compared with Control[HPAP] as demonstrated in the heatmap (scale represents $\log_2$FC) in Fig S5A and B. We observed a subset of inversely correlated DEGs including regulators of insulin secretion such as the subunit of β-cell K[ATP] channel, ATP Binding Cassette Subfamily C Member 8 (*ABCC8*), phosphodiesterase 8B (*PDE8B*), a regulator of cAMP and cGMP degradation, bone morphogenic protein 5 (*BMP5*), which is involved in autophagy, BAI1 Associated Protein 3 (*BAIAP3*), encoding for a $Ca^2$-dependent and RPH3AL Rab GTP effector of late exocytosis, the copper chaperone of superoxide dismutase, and the Regulator Of G Protein Signaling 16 (*RGS16*), all of which were found up-regulated in T2D[HPAP] compared with T2D[PFKFB3] (Fig S5B). Folliculin (*FLCN*) was the only gene found up-regulated in T2D[PFKFB3] and down-regulated in T2D[HPAP] (Fig S5B). FLCN can activate mTORC1 kinase by stimulating GTP hydrolysis of Rag GTPases (Ramirez Reyes et al, 2021). FLCN also plays a role in autophagy and modulation of glycolysis (Ramirez Reyes et al, 2021). Down-regulation of ribosomal genes was coordinated between the individual DEGs (adjusted *P* < 0.05) from Control[HPAP], AAB[HPAP], T2D[HPAP], and T2D[PFKFB3] (Fig S5C and D). The shared and non-shared ribosomal and mito-ribosomal DEGs between the Control[HPAP], AAB[HPAP], T2D[HPAP], and T2D[PFKFB3] are listed in Table S6.

We performed STRING analysis (Szklarczyk et al, 2021a, 2021b) to identify protein–protein interactions based on physical evidence with high edge confidence (>90%) (Szklarczyk et al, 2021b) (Fig 2A–C). We found frameworks of protein associations with clusters visualizing the dominance of ribosomal genes (RiBi) in AAB[HPAP] and T2D[PFKFB3] and mitochondrial respiration with *Cytochrome-c oxidase activity*, *Ubiquinol-cytochrome-c reductase activity*, *Electron transfer activity, and NADH-dehydrogenase (ubiquinone) activity* in T2D[HPAP] datasets (Figs 2A–C and S6A–C). Interestingly, in the mitochondrial respiration framework we identified one cluster related to *Mitochondrial Translation* and *ATP Synthases* both in T2D[HPAP] and in T2D[PFKFB3], in the latter much less present (Fig S6A and B). The connectivity framework of small and large ribosomal subunits was already present in prediabetes (AAB[HPAP]) corroborating ribosomal genes as the origin of the β-cell "loser" status. T2D[PFKFB3] represented an intermediate state between prediabetes (AAB[HPAP]) (where the mitochondrial respiration cluster was omitted) and T2D[HPAP] (with the most prominent mitochondrial respiration cluster). These results suggested that the natural history of the "loser" β-cells ("loser" trajectory) is associated with RiBi, which dominates early and mitochondrial respiration dominating the late diabetes phenotype (Fig 2A, top right panel).

Given that T2D[HPAP] and T2D[PFKFB3] may only differ relative to the active and inactive CFC (protected "loser" β-cells), respectively, the unique clusters to T2D[PFKFB3] may hold a clue to CFC inactivation. STRING analysis revealed down-regulation of human leukocyte antigens (HLA) class I (*HLA-A, HLA-B, HLA-C, HLA-E*) and class II (*HLA-DRB1, HLA-DMB*), *Histones*, and components of the *Trypsinogen pathway* specifically in T2D[PFKFB3] DEGs (Figs 2B and S6B). To corroborate these intriguing observations, we carried out an independent analysis.

## hdWGCNA and WGCNA reveal hub genes related to RiBi in all disease states

We applied the R package high-dimension (hd)WGCNA (Morabito et al, 2023) on HPAP scRNA-seq data for each disease state. We obtained various networks as an output in the form of co-expression modules with the top 10 hub genes. Co-expression modules in each disease condition were correlated with "metacell" states of pancreatic β-cells (Tables S7, S8, S9, and S10). The "metacells" are defined as groups of transcriptionally identical and distinct cell states. They were created using a bootstrapped aggregation (bagging) algorithm by applying the K-nearest neighbors method (Morabito et al, 2023).

To summarize the expression of the entire module into a single metric, module eigengenes (MEs) were created for each metacell population. MEs are defined as the first principal component of the module's gene expression matrix that describes the entire co-expression module. The turquoise co-expression module dominated because of the overt number of co-expressed genes in transcriptionally identical metacells for each disease condition (Fig 3A–J). This was corroborated by the falling dendrograms in which among all disease states the turquoise module was largest in the T2D[PFKFB3] (Fig 3A, C, and E). The gray modules comprised genes that were not grouped into any co-expression module and were therefore excluded from the subsequent analysis. To identify co-expression networks based on the strong gene–gene co-expression without the noise of weak correlations, soft power thresholds of 12, 16, 20, and 14 were selected for mean, median, and maximum (max) connectivity to reach the Scale Free Topology Model Fit greater than 0.8 for Control[HPAP], AAB[HPAP], T1D[HPAP], and T2D[HPAP], respectively (Fig S7). Feature plots of different co-expression modules are presented in Fig S8.

**Figure 1. Elucidating the "loser" β-cell trajectory in diabetes**
**(A)** Scheme of PFKFB3-dependent loss of β-cell function and locked CFC impeding functional β-cell regeneration. Protein aggregates lead to up-regulation of the HIF1α–PFKFB3 pathway that delinks glucose sensing from mitochondrial respiration and insulin secretion. PFKFB3 expression promotes "loser" β-cell survival by CFC deactivation. Targeting PFKFB3 reactivates CFC and clearance of "loser" β-cells. **(B)** Genomic pipeline to reveal the transcriptional profile of "loser" β-cells and PFKFB3-expressing dysfunctional human β-cells across health, prediabetes (AAB), T1D, and T2D using HPAP scRNA-seq database and the spatial transcriptomics from nPOD T2D pancreata, respectively. **(C)** Diagrammatic presentation of the anticipated effect of PFKFB3 targeting on "loser" β-cells in healthy versus diabetic islets. "Loser" (L) and "winner" (W) β-cells are part of CFC, and CFC is active in health. However, in diabetes, "loser" β-cells that express high PFKFB3 (P+L) are no more CFC-elective. When PFKFB3 is targeted, CFC becomes specifically unlocked in P+L cells, whereas β-cells expressing low PFKFB3 (P) are not affected in the absence of L signatures. **(D)** Dot plot showing the coverage of Reactome enriched pathways of DEGs for AAB[HPAP], T2D[HPAP], and T2D[PFKFB3]. **(E)** Overlap of enriched pathways across Control[HPAP], prediabetes (AAB[HPAP]), T2D[HPAP], and T2D[PFKFB3]. Top ten shared and enriched pathways across the Control[HPAP], prediabetes (AAB[HPAP]), T2D[HPAP], and T2D[PFKFB3] are shown as a list on the left bottom corner.

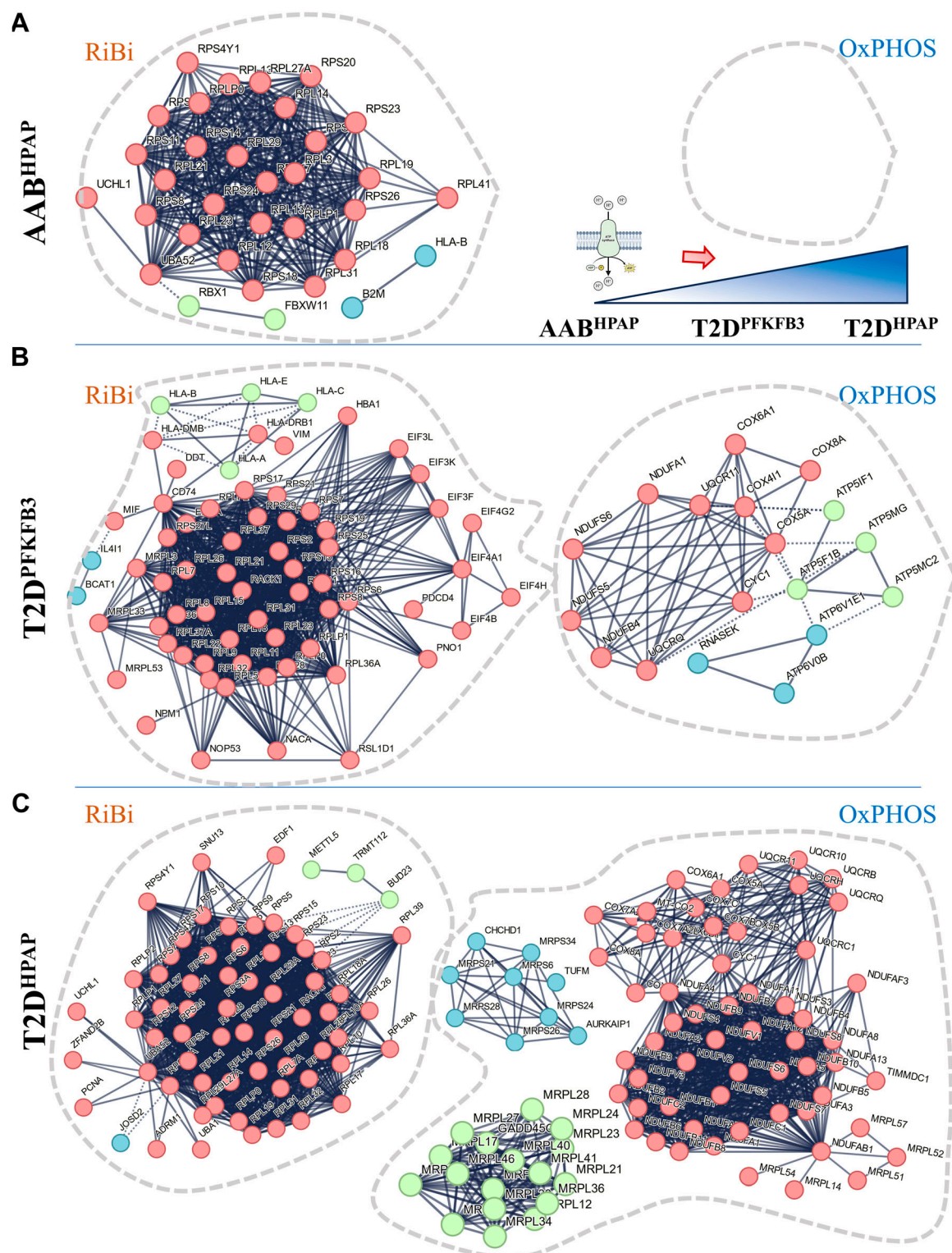

**Figure 2. STRING analysis reveals ribosomes, mitoribosomes, and mitochondrial respiration are at the core of the "loser" signature.**
**(A)** Experimental evidence (>90% confidence level)–based protein–protein interaction framework of DEGs with a cluster of ribosomal biosynthesis and the absence of the cluster for mitochondrial respiration in AAB[HPAP]. The drawing inset explains the growing presence of OxPHOS cluster from prediabetes to sincere T2D. **(B)** Experimental evidence (>90% confidence level)–based protein–protein interaction framework of DEGs with clusters for ribosomal biosynthesis and mitochondrial respiration in T2D[PFKFB3]. **(C)** Experimental evidence (>90% confidence level)–based protein–protein interaction framework of DEGs with clusters for ribosomal biosynthesis and mitochondrial respiration in T2D[HPAP].

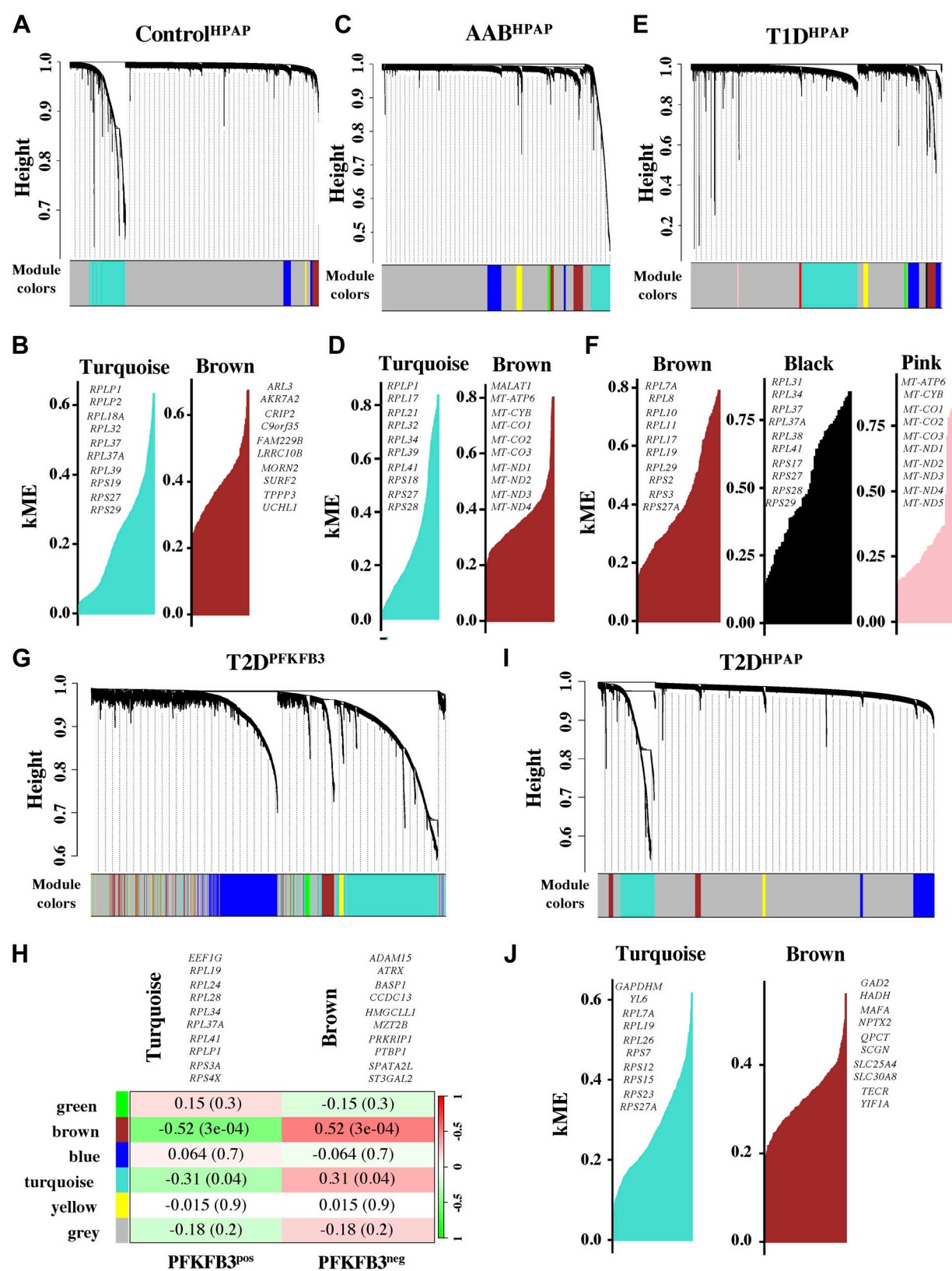

**Figure 3. hdWGCNA and WGCNA reveal hub genes related to RiBi in all disease states.**
**(A, B, C, D, E, F, G, H, I, J)** Falling dendrograms and module eigengenes representing transcriptionally identical "metacell" states in Control[HPAP], (C, D) in AAB[HPAP], (E, F) in T1D[HPAP], (G, H) in PFKFB3-positive β-cells from T2D[PFKFB3], and (I, J) in "loser" β-cells in T2D[HPAP]. **(B, D, F, H, J)** Top 10 hub genes with connectivity constant for module

The eigengene-based connectivity (kME) with each gene in the scRNA-seq data assisted in revealing the hub genes of each module. In Fig 3A, we presented only turquoise and brown modules out of four types of MEs (blue, turquoise, brown, and yellow) found in the Control[HPAP] and they overlapped with the GSEA. The turquoise module eigengenes comprised RiBi-related hub genes such as Ribosomal Protein Lateral Stalk Subunit P1 (RPLP1), and Ribosomal Proteins (RP) L32, RPL37, RPL18A, RPS19, RPLP2, RPL37A, RPS29, RPS27, and RPL39, constituting a shared feature between Control[HPAP] and other datasets. In the AAB[HPAP] dataset, we found turquoise and brown module eigengenes comprised RiBi genes and mitochondrial genes, respectively (Fig 3C and D).

Interestingly, in T1D[HPAP] we found two module eigengenes (brown and black) related to RiBi and pink module eigengenes for the mitochondrial hub genes. This contrasted the results referring to DEGs derived from "loser" criteria in T1D. Thus, T1D unlike other disease states may hold different "loser" criteria for selection of ribosomal and mitochondrial hub genes (Fig 3E and F). Surprisingly, mitochondrial modules were found in AAB[HPAP] and T1D[HPAP], different from the GSEA (Fig 3D and F). Because hdWGCNA is computed from the normalized gene expression matrix to generate metacell expression matrices using the K-nearest neighbors parameter, hub genes in MEs might not necessarily overlap with DEGs from the scRNA-seq analysis.

To be able to compare the MEs between T2D[PFKFB3] and T2D[HPAP], we used conventional WGCNA (Langfelder & Horvath, 2008) for analysis of the matrix from spatial transcriptomics. We identified brown and turquoise modules in T2D[PFKFB3] comprising genes involved in β-cell function (P < 0.05) and RiBi hub genes (P < 0.05), respectively (Fig 3G and H). Similar to T2D[PFKFB3], in T2D[HPAP] we found a dominating turquoise ME in reference to RiBi subunits together with Glyceraldehyde-3-Phosphate Dehydrogenase (GAPDH) and Myosin Light Chain 6 (MYL6) but not the module eigengenes for the mitochondrial hub genes, which was different from the results of the GSEA from T2D[HPAP] (Fig 3I and J). Thus, collectively, the turquoise module for RiBi hub genes was shared across all datasets and was not correlated to the module represented by mitochondrial hub genes (Fig S9).

## Key transcriptional drivers of "loser" signature in β-cells

We used Mergeomics weighted Key Driver Analysis to determine the key drivers of the differential gene expression in disease by applying gene network topology and edge weight information from DEGs (see the Materials and Methods section) (Fig 4A–F). Herein, we identified networks of functionally related candidate hub genes with regulatory roles in the disease gene expression networks. In AAB[HPAP], key drivers were represented by genes playing an important role in the elongation step of protein synthesis, Eukaryotic Translation Elongation Factor 2 (EEF2), and Eukaryotic Translation Initiation Factor 2 Subunit Alpha (EIF2S1, RPL26, RPL4, and RPLS1)

with Eukaryotic Translation Elongation Factor 2 (EEF2, RPL4, and RPLS1) making the functional connection (Fig 4A and B).

T2D[PFKFB3] and T2D[HPAP] shared a key driver gene, the Signal Recognition Particle 19 (SRP19). Other key drivers in T2D[PFKFB3] included Ribosomal Protein L22 Like 1 (RPL22L1), and the structural constituent of ribosome and SERPINE1 mRNA Binding Protein 1 (SERBP1) involved in SERPINE1 mRNA stability and ribosome hibernation, a process during which ribosomes are prevented from proteasomal degradation (Shetty et al, 2023). We also identified Signal Peptidase Complex Subunit 1 (SPCS1) that catalyzes the cleavage of nascent proteins during their translocation in ER (Liaci et al, 2021), and Tumor Protein Translationally Controlled 1 (TPT1) (Chen et al, 2013). Upon glucose stimulation, TPT1 is translocated to the mitochondria and the nucleus. Gene knockdown of TPT1 induces apoptosis, and its overexpression reduces ER-stress–related apoptosis (Diraison et al, 2011) (Fig 4C and D). Similar to TPT1, SERBP1-dependent fatty acid synthesis is required for insulin secretion at high glucose concentrations (Diraison et al, 2008). The T2D[PFKFB3] key drivers RPL22L1, SERBP1, and TPT1 formed functional connections and are all related to increased resistance to stress (RPL22L1 and SERBP1 partaking in ribosome hibernation [Shetty et al, 2023], and TPT1 in resistance to oxidative and ER stress [Diraison et al, 2011; Chen et al, 2013]).

In T2D[HPAP], key drivers were presented by Mitochondrial Ribosomal Protein S7 (MRPS7) from mitochondrial protein synthesis; SEC61 Translocon Subunit Alpha 2 (SEC61A2), a signal peptide–containing precursor for co-translational translocation of nascent polypeptides across the ER, forming an ER ribosome receptor and a gated pore; Signal Sequence Receptor Subunit 4 (SSR4) binding calcium and regulating ER-resident proteins; SEC61 Translocon Subunit Gamma (SEC61G) with ATPase activity; and SRP19 enabling 7S RNA binding activity. The functionally connected key drivers in T2D[HPAP] comprised SEC61 subunits and SSR4, falling under mitochondrial and ER regulatory genes (Fig 4E and F).

Collectively, these analyses confirmed that PFKFB3-positive β-cells are cells with "loser" signatures that show reduced HLA expression, evidence of potential escape from immunosurveillance (MacLean et al, 2010; Perfeito et al, 2014; Schoustra et al, 2016). Because PFKFB3-positive "loser" β-cells are long-lived cells carried on from prediabetes, we asked about the mechanism by which PFKFB3 can exert protection despite the global distortion of RiBi and mitochondrial respiration.

## SNP-SNP analysis reveals PFKFB3 *rs1983890* interplay with the anti-apoptotic gene *MAIP1*

The positive epistasis as an epiphenomenon that can counter the global distortion of RiBi has been described previously in yeast (Tutaj et al, 2023), and we set to investigate it in relation to PFKFB3 gene–gene interactions (GGI). We analyzed interactions between the PFKFB3 polymorphic allele on the chromosome 10p15.1 locus

eigengenes (kME) of each hub gene showing dominant turquoise (RiBi) and brown (OxPHOS) modules for (B) Control[HPAP] and (D) AAB[HPAP]; (F) dominant brown and black (RiBi), and pink (OxPHOS) modules for T1D[HPAP]; (H) dominant turquoise (RiBi) and brown (OxPHOS) modules in T2D[PFKFB3] with marked Z-score and P-values; and (J) dominant turquoise (RiBi) and brown (OxPHOS) modules with kME value in T2D[HPAP]. Each leaf on the dendrogram represents a single gene, and the color at the bottom indicates the co-expression module assignment.

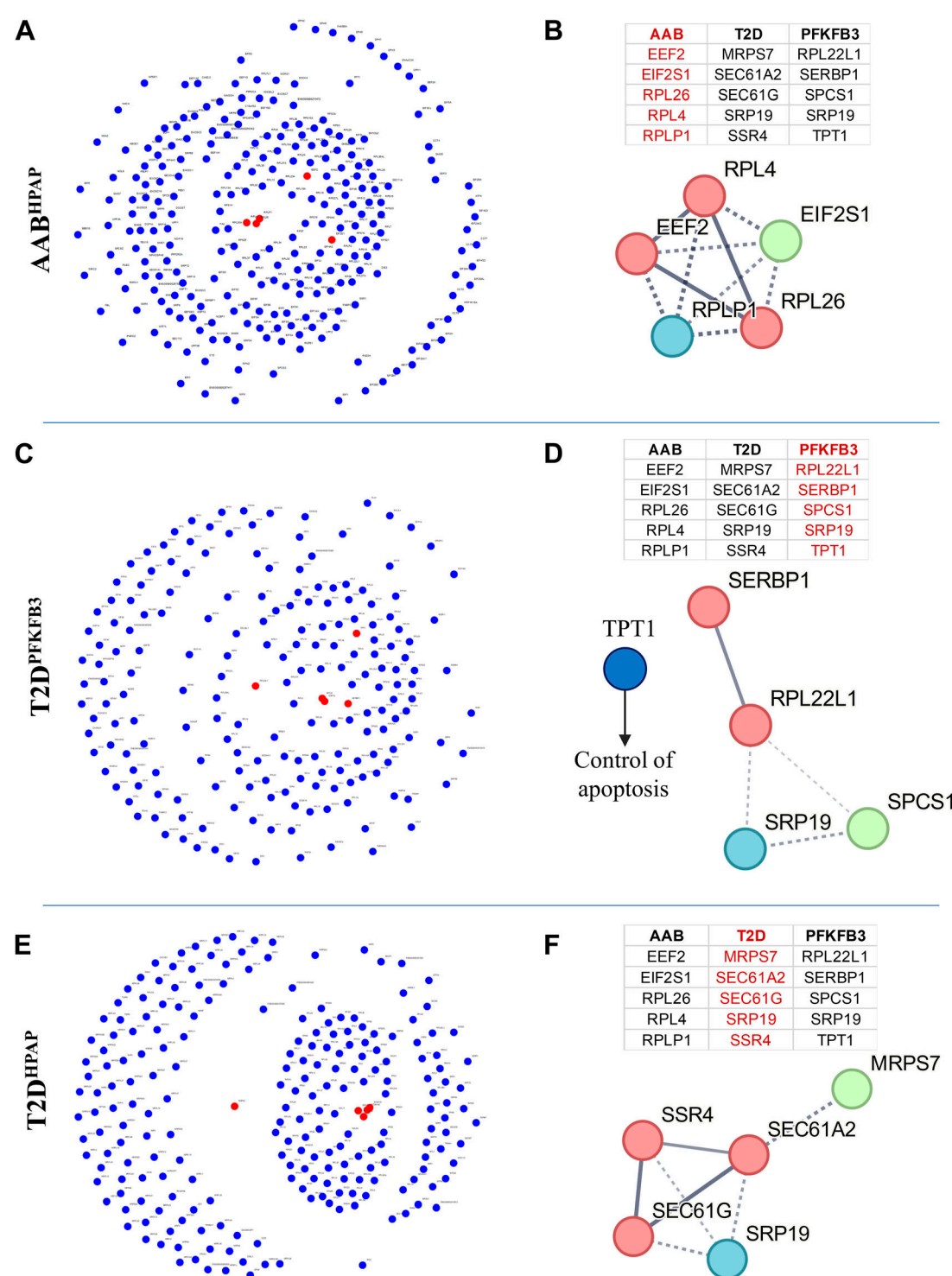

**Figure 4. Key transcriptional drivers of "loser" signature in β-cells.**
**(A, B, C, D, E, F)** Network of key driver hub genes (presented as red nodes and shared genes as blue nodes) with a list of top five key driver genes (highlighted in red) in the adjacent table and the respective protein–protein interaction network in AAB[HPAP], (C, D) in T2D[PFKFB3], (E, F) in T2D[HPAP]. Please note that TPT1 anti-apoptotic gene is not part of the depicted cluster in (D).

(*rs1983890*) (Wallace et al, 2015) and all annotated SNPs from the HPAP dbGAP corresponding to the list of DEGs in PFKFB3-positive β-cells (Fig 5A). Using case (all diabetic donors) to control (non-

diabetic donors) approach, we found only two genes with significant (adjusted $P < 0.05$) SNP interactions, the matrix-AAA peptidase interacting protein 1 (*MAIP1*) (adj. $P = 0.0113$); and the ArfGAP With

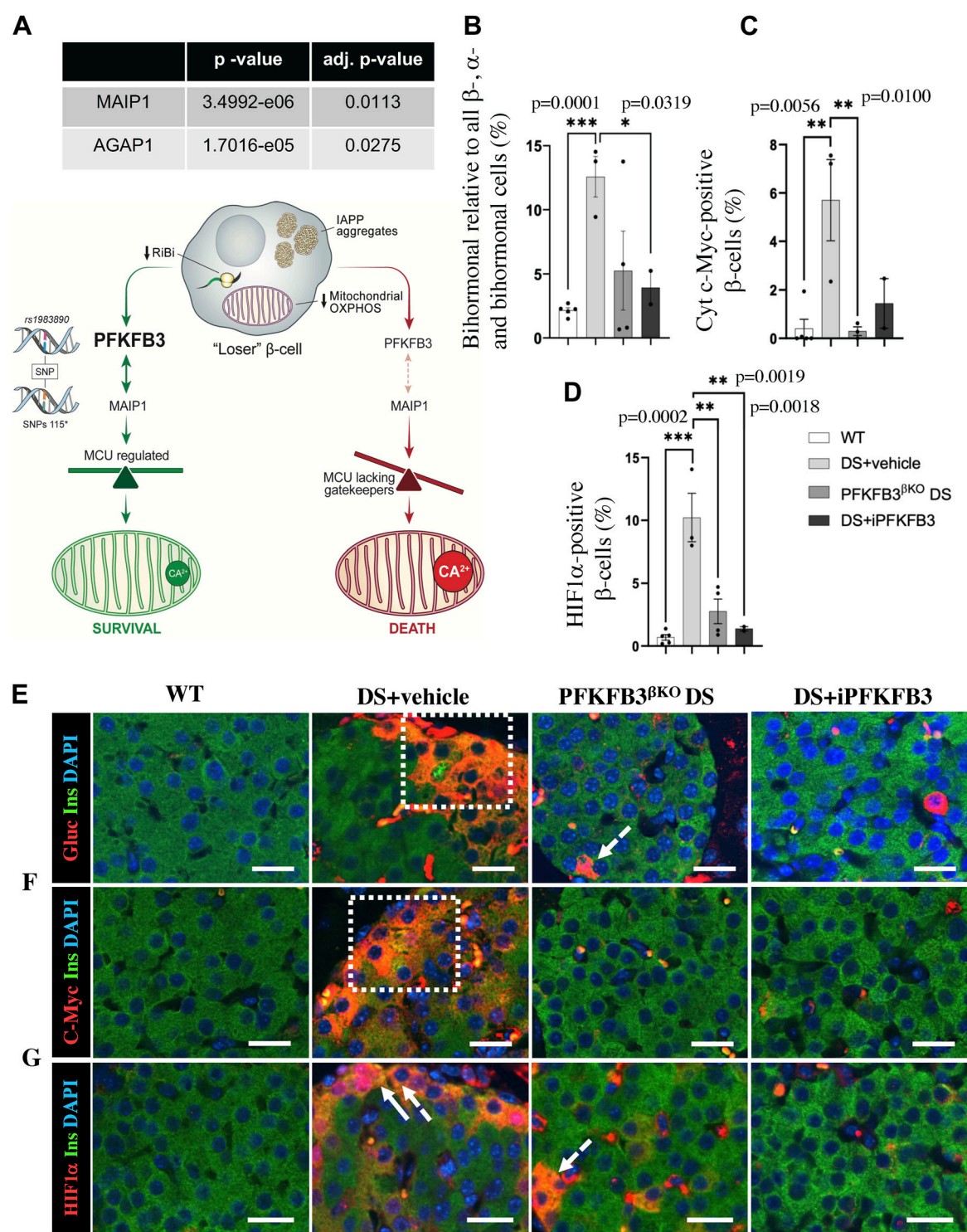

**Figure 5. PFKFB3 inhibition eliminates dysfunctional β-cells that are protected by positive epistasis.**

**(A)** PFKFB3 rs1983890-SNP interaction analysis with the DEGs in PFKFB3-positive relative to PFKFB3-negative β-cells revealed by spatial transcriptomics yielded significant interactions with collective SNPs of AGAP1 and MAIP1 (SNPs 115). Diagrammatic reconstruction of the results indicates a proposed mechanism of PFKFB3-mediated survival of injured "loser" β-cells. MAIP1 is an anti-apoptotic gene that controls mitochondrial Ca²⁺ via MCU uniporter. MCU is composed of channel-forming subunits, EMRE, and a gatekeeper subunit MICU. The gating of the MCU complex requires an association between EMRE and MICU. MAIP1 may protect EMRE from i-AAA–mediated degradation (Opalinska & Janska, 2018) preventing constitutive activation of MCU and cell death. **(B)** Quantitative representation of insulin-positive and glucagon-positive (bihormonal) injured cells (% of all β-, α-, and bihormonal cells). **(C)** Quantitative representation of c-Myc–positive injured β-cells (% of all β-cells). **(D)** Quantitative representation of HIF1α-positive injured β-cells (% of all β-cells). **(E, F, G)** Representative immunofluorescence images of islets, from wild-type mice exposed to high-fat diet, HFD (WT), mice under diabetogenic stress treated with vehicle (IAPP+HFD = DS; DS + vehicle), mice with conditional PFKFB3 knockout exposed to

GTPase Domain, Ankyrin Repeat And PH Domain 1 (*AGAP1*) (adj. *P* = 0.0275), presented in the Tables S11 and S12. MAIP1 interacts with the m-AAA peptidase, preventing degradation of the essential mitochondrial calcium uniporter (MCU) regulator (EMRE).

These results indicated that PFKFB3 may play via MAIP1 a direct role in the survival and protection of "loser" β-cells and that targeting PFKFB3 can facilitate the clearance of these cells.

Therefore, we set out to analyze the phenotypic consequences of PFKFB3 depletion or inhibition in mice undergoing diabetogenic stress and the feasibility of targeting PFKFB3 at a systemic level.

Imaging analysis of the immunostainings from the h-βTG mouse pancreata from DS+vehicle compared with the WT controls indicated accumulation of different types of injured β-cells such as double insulin- and glucagon (bihormonal)-, cytoplasmic c-Myc–positive, and HIF1α-positive β-cells (Fig 5B–G). We have not found a transcriptional up-regulation of PFKFB3 in "loser" β-cells. However, PFKFB3 could be stabilized in a post-translational fashion implicating APC/Cdh1 and Emi1 (Tudzarova et al, 2011; Cappell et al, 2018). In that sense, we found a down-regulation of APC/Cdh1 (*FZR1*) in T2D$^{HPAP}$ (log$_2$FC = −0.21; *P* = 0.004), whereas in T2D$^{PFKFB3}$, the down-regulation was not significant (log$_2$FC = −0.004; *P* = 0.6) (Table S13).

PFKFB3-expressing β-cells exceeded the number of individual injured β-cells, and PFKFB3 knockout led to the clearance of all injured β-cells (Min et al, 2022). We separated the PFKFB3 targeting group (n = 5) into three animals subjugated to β-cell–specific knockout of PFKFB3 (PFKFB3$^{βKO}$ DS) as a positive control (Min et al, 2022) and two animals that received the PFKFB3 inhibitor AZ67. PFKFB3$^{βKO}$ DS led expectedly to the diminishment of all types of injured β-cells (Figs 5B–G, S10, S11, and S12). 7 wk of i.p. administration of the PFKFB3 inhibitor AZ67 (DS+iPFKFB3 group) recapitulated the effect of PFKFB3$^{βKO}$ DS as confirmed with quantification of all injury markers positive versus all β-cells (Fig 5B–G). Bihormonal cells (that are increased in high-fat diet [HFD]), HIF1α-positive (dysfunctional β-cells reflecting islet inflammation induced by HFD), and cytoplasmic c-Myc–positive β-cells (indicators of cells undergoing hIAPP-induced calpain activation and toxicity), which were significantly up-regulated in DS+vehicle, were depleted in DS+iPFKFB3 mice (Fig 5B–G).

To understand the relevance of MAIP1 expression in stressed β-cells, we immunostained serial sections of the whole pancreas from WT control, DS+vehicle, and DS+iPFKFB3 mice with MAIP1 antibody and PFKFB3 antibody followed by a comparison of the identical islets. We counted β-cells from all the islets in the whole pancreas section.

High MAIP1- and PFKFB3-immunopositive β-cells in DS+vehicle mice co-localized and overlapped in frequency showing 24.4% ± 2.2% and 29.6% ± 3.1% SEM of all β-cells, respectively, marking β-cells with reduced insulin expression (Figs 6A–H and I, S13, and S14). The β-cells with an overlap in high PFKFB3 and MAIP1

expression are shown as a subset in Figs 6G and H and S14. The immunopositivity of both MAIP1 and PFKFB3 was below 1% in WT+HFD and DS+iPFKFB3 pancreata (Fig 6I). Given that PFKFB3 inhibition cannot affect the PFKFB3-MAIP1 gene–gene interaction, depletion of MAIP1-positive β-cells supports the idea of clearance over β-cell restoration. Islet-specific MAIP1 staining was further confirmed in the human nPOD sections from the two T2D donors (#6255 and #6186) used in the spatial transcriptomics and one independent T2D nPOD donor (#6300) (Fig 6J–L). As non-diabetic controls, we immunostained nPOD donor IDs #6091, #6048, and #6096. The strongest MAIP1 immunopositivity was associated with β-cells with reduced insulin expression and accounted to 36.5% ± 7.4% SEM of all β-cells in T2D (Fig 6F). Unlike MAIP1, AGAP1 up-regulation was observed in a subset of non–insulin-positive islet cells (Fig S15) in line with the Human Protein Atlas (https://www.proteinatlas.org/ENSG00000157985-AGAP1/tissue+cell+type) showing AGAP1 expression in endothelial cells, fibroblasts, macrophages, and ductal and T cells but no expression in β-cells. When normalized with all islet non–insulin-positive cells, AGAP1 expression pertained to comparable 29.2% ± 3.6% SEM, 29.3% ± 2.4% SEM, and 27.4% ± 3.4% SEM in WT+HFD, DS+vehicle, and DS+iPFKFB3, respectively (Figs 6M and S15).

We asked whether clearance of injured "loser" β-cells under DS conditions would be sufficient to restore metabolic control in the DS mice.

## Systemic PFKFB3 inhibition improves metabolic performance of h-βTG (T2D) mice

We administered the PFKFB3 inhibitor AZ67 daily (at 28 mg/kg body weight) from day 1 of exposure to HFD for 7 wk. We performed intraperitoneal glucose tolerance test (IP-GTT) at 4 and 6 wk of iPFKFB3 treatment and an insulin tolerance test at 5 and 7 wk of iPFKFB3 treatment. The experimental timeline of PFKFB3 inhibitor (iPFKFB3) administration and metabolic measurements are depicted in Fig 7A.

The clearance of injured β-cells by PFKFB3 inhibition correlated with the improved glucose tolerance as demonstrated by IP-GTT and by AUC of blood glucose (±SEM, *P* < 0.05) after 4 wk (n = 7–8, ±SEM, *P* < 0.05) (Fig 7B, D, and E) and 6 wk (two independent experiments, n = 4–8/group, ±SEM, *P* < 0.05) (Fig 7C, G, and H). Improved glucose tolerance led to reduced fasting blood glucose (Fig 7E and H). Moreover, insulin sensitivity was increased although not reaching significance in DS+iPFKFB3 mice after 5 wk (n = 7–8, ±SEM, *P* < 0.05) (Fig 7F) and 7 wk (two independent experiments, n = 4–8/group, ±SEM, *P* < 0.05) (Fig 7I) when compared to the DS+vehicle group.

Collectively, these data indicated that targeting PFKFB3-positive β-cells with "loser" signature by PFKFB3 inhibition leads to

diabetogenic stress for 6 wk (PFKFB3$^{βKO}$ DS), and mice under diabetogenic stress receiving the PFKFB3 inhibitor AZ67 for 6 wk (DS + iPFKFB3); (E) double immunostained for insulin and glucagon (to reveal bihormonal cells), (F) immunostained for cytoplasmic c-Myc, and (G) HIF1α (all markers in red), insulin (green), and nuclei (blue) showing eradication of marker-positive injured β-cells in PFKFB3$^{βKO}$ DS and DS + iPFKFB3 mice. The sectioning was performed across the whole pancreas. For quantification of the markers' frequency, the whole pancreas section per mouse from the treatment group was used. We have counted all the pancreatic islets from each whole pancreas section (150–200 islets per treatment group) (n = 3). Scale bar: 50 μm.

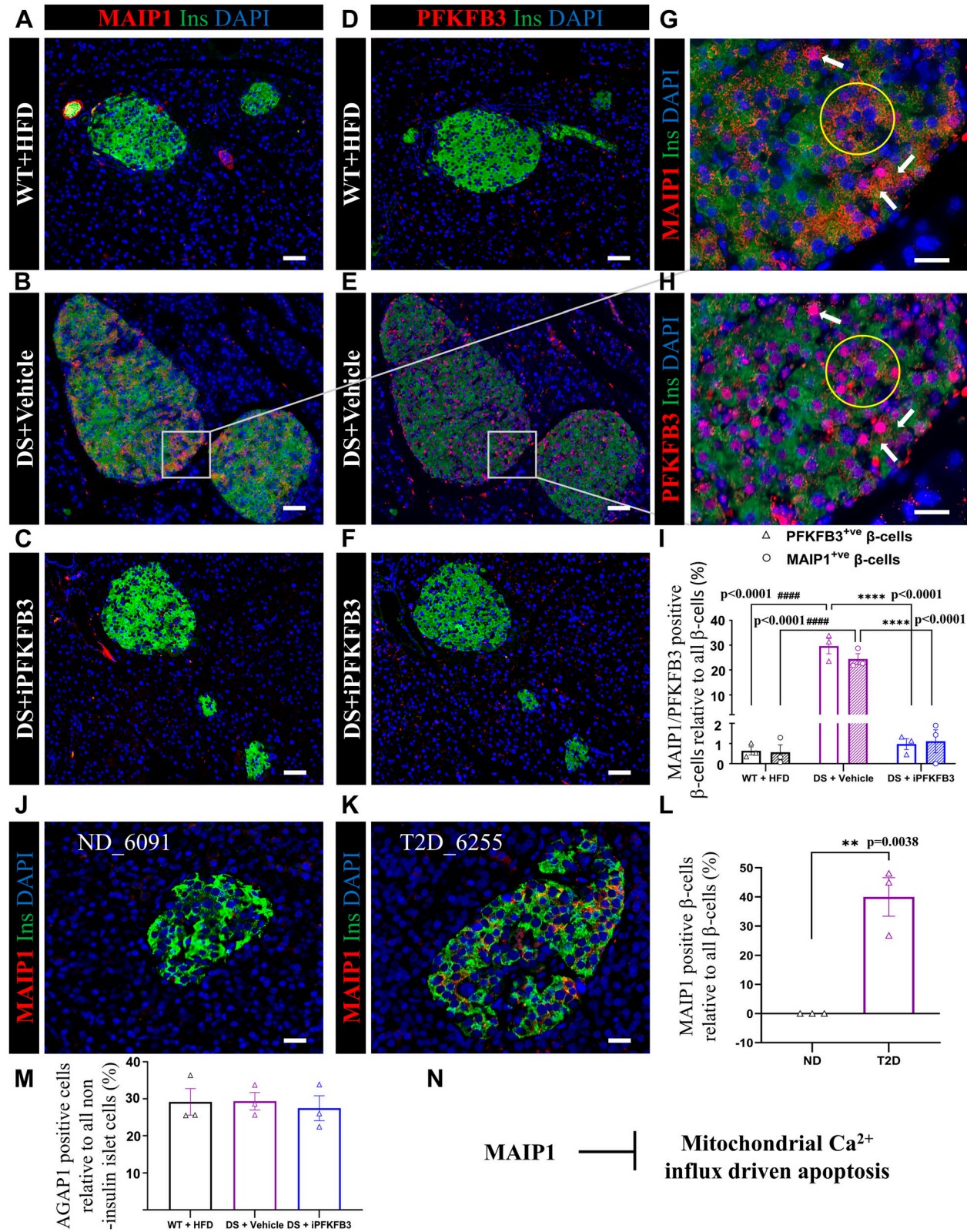

**Figure 6. Overlap in β-cells with high PFKFB3 and MAIP1 expression.**
**(A, B, C)** Representative immunofluorescence images of islets from mice as described in Fig 5, double immunostained for insulin and MAIP1 (red), insulin (green), and nuclei (blue). The sectioning was performed across the whole pancreas. For quantification of MAIP1 frequency, we have counted all the pancreatic islets from each whole pancreas section, that is, 122–201 islets per treatment group (n = 3). Scale bar: 50 μm; **(D, E, F)** Immunostained for insulin, and PFKFB3 (red), insulin (green), and nuclei (blue).

diminishment of dysfunctional "loser" β-cells, restoring glucose tolerance in the human-like model of T2D.

# Discussion

CFC has been successfully validated in post-mitotic cells (Coelho & Moreno, 2019; Coelho et al, 2018; Gradeci et al, 2021; Vieira et al, 2024), and here, we evaluated the hypothesis that "loser" and "winner" cell imbalance may underlie one of the pathogenic mechanisms in T2D.

We demonstrated that PFKFB3-positive β-cells are "loser" cells of CFC and that targeting PFKFB3 can facilitate "loser" β-cell clearance by reactivation of CFC. β-Cell "loser" signature emerged from the high level of transcriptional convergence between AAB[HPAP], T2D[HPAP], and T2D[PFKFB3]. Although we have not found transcriptional up-regulation of PFKFB3 in the "loser" HPAP datasets, we found that APC/Cdh1 E3 ubiquitin ligase that regulates PFKFB3 stability (Tudzarova et al, 2011) was down-regulated in T2D[HPAP], in line with PFKFB3 protein up-regulation in diabetes (Wigger et al, 2021). PFKFB3-positive β-cells in T2D shared a global stoichiometric distortion of RiBi with "loser" β-cells in AAB[HPAP] and T2D[HPAP]. Both RiBi and mitochondrial genes were enriched in the GSEA and metacells from WGCNA, with global RiBi down-regulation representing a measurable metacell entity across all β-cells from pre-diabetes to sincere T2D. These findings are interesting because originally the "loser" status was demonstrated by one dominant ribosomal subunit (Minute) mutation in *Drosophila* (Carnegie Institution of Washington, 1923). The enrichment showing the distortion of RiBi in Control[HPAP] has not reached significance, which may be due to an active CFC in health. Although the overlap existed between β-cells from Control[HPAP], AAB[HPAP], T2D[HPAP], and T2D[PFKFB3], there was no shared GSEA signature with T1D. Interestingly, met-acells in T1D were represented by both ribosomal and mitochondrial hub genes, thereby not correlating with the GSEA results. The lack of correlation may point to different CFC mechanism(s) and/or enhanced attrition of "loser" β-cells by autoimmunity in T1D. T2D[HPAP] GSEA was dominated by the down-regulation of mito-chondrial respiration genes. T2D[PFKFB3], which pertained to the global shutdown of RiBi, had yet only one enriched category of genes encoding mitochondrial respiration, positioning itself be-tween AAB[HPAP] and T2D[HPAP] as shown in the heatmaps (Fig S5A, C, and D). Interestingly, the emerging difference between AAB[HPAP], T2D[PFKFB3], and T2D[HPAP] was also corroborated by down-regulation of RPS12 in T2D. Down-regulation of RPS12 (Ji et al, 2019) shared by T2D[PFKFB3] and T2D[HPAP] mimics phenotypically the missense

mutations of RPS12, which can prevent attrition of T2D "loser" cells by CFC (Kale et al, 2018).

PFKFB3-positive "loser" β-cells were dominated by GO terms that pertain to mRNA and protein quality control during translation at the ribosome and/or ribosome hibernation (*RPL22L1*, *SERBP1*, and *TPT1*), transcriptional down-regulation of cap-dependent and cap-independent translational machinery, and stalling of the initiation step (Richter & Coller, 2015) complementing transcriptional suppression.

The profound down-regulation of RiBi and genes of mitochon-drial respiration in "loser" β-cells indicates that conserving energy by shutting down the energy-expensive process of making proteins is a priority of long-lived dysfunctional β-cells. This could be, for example, critical for preserving proteosynthetic machinery and orphaned ribosomal proteins for an intact restart of RiBi once the stress is resolved (Buchan & Parker, 2009; Wheeler et al, 2016). Why is the energy conservation necessary in "loser" β-cells? Previously, it was demonstrated that down-regulation of RiBi or metabolism reduction rescued lethal *Drosophila* mutants during development because of redundancy of repressor activities under those con-ditions (Cassidy et al, 2019). Therefore, strained transcriptional regulation and genomic stability under stress can be protected from deleterious consequences under energy conservation. The broader benefit of energy conservation known as hypometabolism on an organismal level under environmental constraints was demonstrated before with torpor or hibernation (Giroud et al, 2020) and is currently considered as a radiation injury–preventative approach for the human space flights (Ghosh et al, 2017).

We reasoned that breaking down T2D[PFKFB3] unique genetic makeup that differs from T2D[HPAP] might offer a clue to the specific role of PFKFB3 in the long-term survival of "loser" β-cells in T2D. One of the most remarkable features of T2D[PFKFB3] DEGs was the down-regulation of HLA class I and II (Drake et al, 2006; Adachi et al, 2022). This is in line with the reduced haplotype frequency of HLA co-occurring with PFKFB3 polymorphism (Chen & Chen, 2019). The unique HLA makeup in T2D[PFKFB3] poses an intriguing possibility that immunity plays a role in the removal of PFKFB3-positive "loser" β-cells (Johnston, 2009). Down-regulation of HLA is connected to a fail-safe response by natural killer (NK) cells (Shi et al, 2011). Nevertheless, chronic in-flammation in diabetes suppresses NK cells allowing potentially PFKFB3 "loser" β-cells to bypass the immunosurveillance.

Another mechanism to explain the prolonged survival of the PFKFB3-positive "loser" β-cells may involve epistasis and the variations in disease outcome (Cordell, 2002). The diabetes sig-nificant (Chen & Chen, 2019) PFKFB3 SNP at the 10p15.1 locus (*rs1983890*) (Wallace et al, 2015) interacted with *MAIP1* and *AGAP1* collective SNPs (Fig 5A) in GGI analysis. The *AGAP1* function is linked

---

**(A, C, D, F)** are showing eradication of MAIP1- and PFKFB3-positive β-cells in WT+HFD and DS+iPFKFB3 mice, respectively. The sectioning was performed across the whole pancreas. For quantification of PFKFB3 frequency, we have counted all the pancreatic islets from each whole pancreas section, that is, 170–191 islets per treatment group (n = 3). Scale bar: 50 μm. **(G, H)** Insets from identical serial sections showing MAIP1 and high PFKFB3 immunopositivity overlap. White arrows indicate β-cells with PFKFB3 and MAIP1 high expression overlap. Scale bar: 20 μm. **(J, K)** Representative immunofluorescence images of islets from human (J) non-diabetic (ND) and (K) T2D donors immunostained for insulin and MAIP1. For quantification of MAIP1 frequency, we have counted all the pancreatic islets from each whole pancreas section, that is, 171–200 islets per donor group (n = 3). Scale bar: 50 μm. **(I)** Quantitative representation of MAIP1 (circles)- and PFKFB3 (triangles)-positive β-cells (% of all β-cells) in the treatment groups described in Fig 5E–G. Data are represented as the mean ± SEM, n = 3, $P < 0.05$. **(L)** Quantitative representation of MAIP1-positive β-cells (% of all β-cells) in human non-diabetic (ND) and T2D donors. n = 3 from each donor group; data are represented as the mean ± SEM, $P < 0.05$. **(M)** Quantitative representation of AGAP1 non–insulin-positive islet cells (% of all non–insulin-positive islet cells) in treatment groups previously described. n = 3; data are represented as the mean ± SEM, $P < 0.05$. **(N)** Flowchart depicting the key role of MAIP1 in the inhibition of mitochondrial-driven apoptosis.

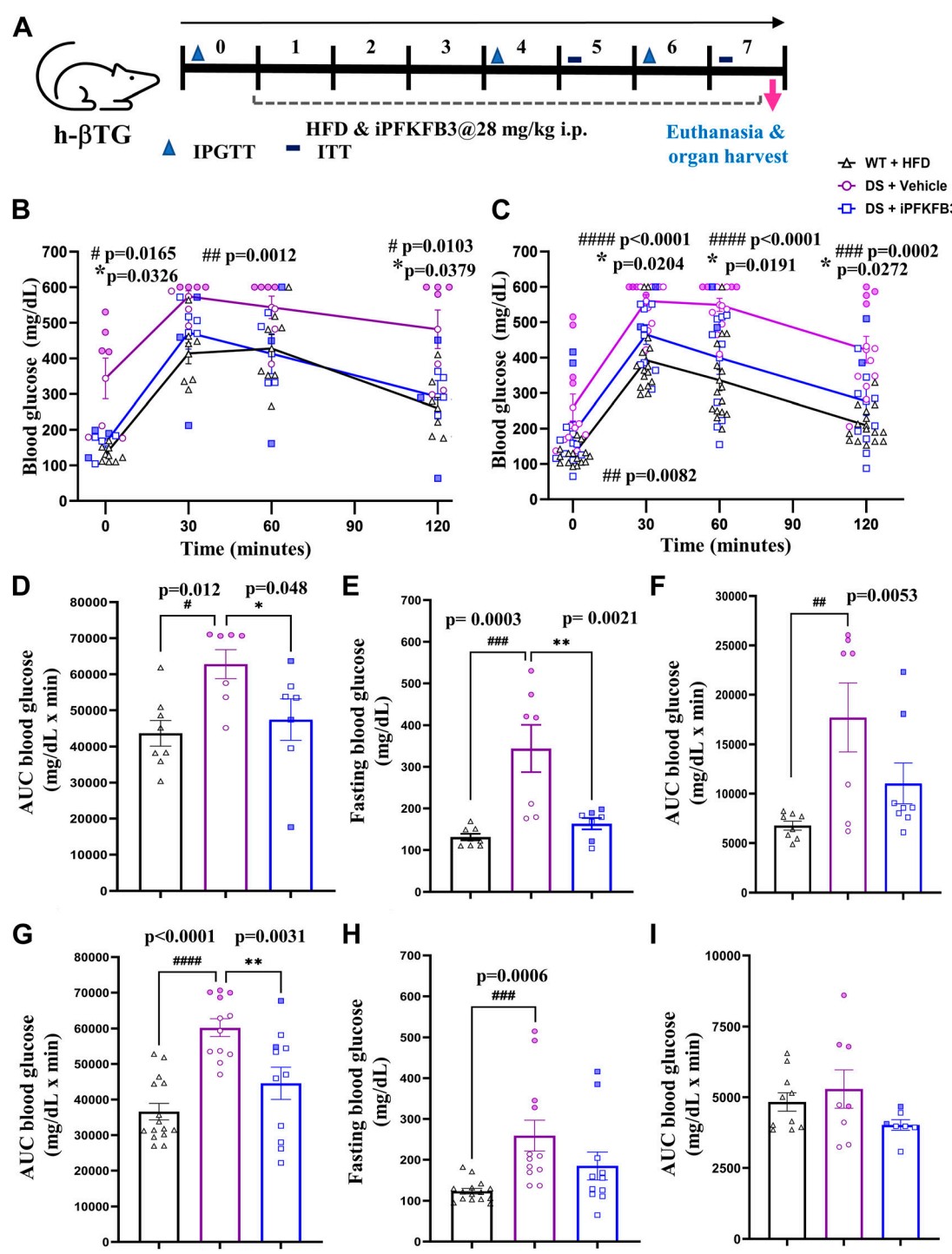

**Figure 7. Systemic PFKFB3 inhibition (iPFKFB3) improves metabolic performance of h-βTG (T2D) mice.**
**(A)** Experimental scheme of PFKFB3 inhibitor treatment of h-βTG mouse; PFKFB3 inhibitor was administered at 28-mg/kg dose via an intraperitoneal route for 7 wk, and metabolic tests were performed (IP-GTT and insulin tolerance test) at indicated time points. **(B, C)** Intraperitoneal glucose tolerance test (IP-GTT) (B) at 4 wk and (C) at 6 wk showed significant improvement in glucose tolerance in the DS + iPFKFB3 as compared to DS + vehicle mice. **(D, E, F, G, H, I)** Comparison of (D) the area under the curve (AUC) from IP-GTT and (E) fasting blood glucose at 4 wk; and (G) the area under the curve (AUC) from IP-GTT and (H) fasting blood glucose at 6 wk showing significant improvement in DS + iPFKFB3 as compared to DS + vehicle mice; AUC from an insulin tolerance test (F) at 5 and (I) 7 wk showing a trend of improvement in insulin sensitivity in DS + iPFKFB3 mice as compared to DS + vehicle mice without reaching significance: n = 8 or 7, n = 7, and n = 7 biological replicates for WT+HFD, DS + vehicle, and DS + iPFKFB3, respectively; statistical testing was performed between # WT versus DS+vehicle, and *DS+iPFKFB3 versus DS+vehicle. At 6 wk, n = 15, n = 12, and n = 11 biological replicates from n = 2 independent experiments for WT+HFD, DS + vehicle, and DS + iPFKFB3, respectively. Data are represented as the mean ± SEM, *P* < 0.05.

to endosome protein trafficking and cytoskeleton, and AGAP1-deficient cells are susceptible to a variety of stressors (Lewis et al, 2023). AGAP1 immunopositivity marked exclusively non–insulin-positive islet cells and did not vary between the groups. Therefore, we focused on the novel MAIP1 protein because of its association with $Ca^{2+}$ homeostasis (Fig 6N) in diabetic β-cells, which is the convergence point of diabetic stressors. MAIP1 exerts an anti-apoptotic function via the protective assembly of the mitochondrial $Ca^{2+}$ uniporter (MCU) (Konig et al, 2016). Interestingly, MAIP1 immunopositivity marked specifically β-cells in DS+vehicle and human T2D donors and its sharp reduction below 1% upon PFKFB3 inhibition in h-βTG mice indicated a potential elimination of MAIP1-positive β-cells connected to the treatment (Huang et al, 2010).

CFC provides a context-dependent removal of injured or dysfunctional cells that has been proven successful in the clinical translation of "synthetic lethality" and immune checkpoint inhibition (Lucchesi, 1968; O'Neil et al, 2017; Shiravand et al, 2022). To unlock CFC and demonstrate feasibility of PFKFB3 targeting at the systemic level, we used the small molecule PFKFB3 inhibitor AZ67 (Boyd et al, 2015). We successfully reduced injured β-cells recapitulating PFKFB3$^{βKO}$ DS, which led to improvement in glucose tolerance. Our study would benefit from real-time monitoring of injured β-cell clearance in in vivo–transplanted human islets, which is presently a main limitation in assessing the temporal nature and functional cell regeneration when attempting reactivation of CFC.

Collectively, we reveal in this study that the RiBi and mitochondrial respiration (cellular energy conservation) represent a "global" phenotypic interface of β-cell fitness that in the presence of PFKFB3 can create an epiphenomenon of positive epistasis via *MAIP1*, whereas reduced HLA expression may help bypass immunosurveillance. Positive epistasis is necessary to suppress CFC favoring the survival and accumulation of dysfunctional β-cells. Selective clearance of dysfunctional "loser" β-cells agnostic of their origin (e.g., senescent cells) by reactivation of physiologically competent CFC is an original approach. This study predicts that T2D patients without PFKFB3 polymorphism should have a better clinical outcome, what remains to be addressed in the future. Pharmacological targeting of PFKFB3 to abort survival of dysfunctional "loser" β-cells holds a promise to change and prevent deterioration of early T2D trajectory given that at this stage both the prevalence of "loser" β-cells and "loser" β-cell survival selectively depend on PFKFB3. The therapeutic ramification of CFC in T2D is emerging with a potential to modify this disease by enrichment with functional β-cells at early onset where β-cell mass is still conserved. This approach when therapeutically developed may find application in other age-related diseases and aging itself, given the shared feature of dysfunctional cell accumulation across many tissues in addition to the pancreas.

# Materials and Methods

### Study design

We analyzed large-scale transcriptomic data of human pancreatic β-cells from two independent datasets. We used the Human Pancreas Analysis Program (HPAP) (Kaestner et al, 2019). HPAP is part of the Human Islet Research Network supported by the National Institute of Diabetes and Digestive and Kidney Diseases (NIDDK), which leverages deep phenotyping of the human endocrine pancreas, thereby accumulating, analyzing, and distributing high-value datasets to the diabetes research community through the HPAP-PANC-DB database. We also used formalin-fixed and paraffin-embedded pancreata from three T2D donors from the Network of Pancreatic Organ Donors with Diabetes (nPOD) program (Campbell-Thompson et al, 2012).

We analyzed scRNA-seq from 31 non-diabetics (Control$^{HPAP}$), 10 prediabetics with or without dysglycemia classified based on 2 or more autoantibodies (AAB$^{HPAP}$), 9 donors with type 1 diabetes (T1D$^{HPAP}$), and 17 donors with type 2 diabetes (T2D$^{HPAP}$) using HPAP annotation of pancreatic β-cells. For each condition, and based on the module score reflecting the expression ($Log_2FC > 0.1$) or not of the "loser genes" *DDIT3*, *Atf3*, *Ppp1r15a*, *RICTOR*, and *Nfe2l2* ($Log_2FC <$ or $= 0$), cells were split into "*loser signature*"–positive and "*loser signature*"–negative β-cells for differential expression analysis between the two groups (Ortiz-Barahona et al, 2010; Valvona et al, 2016). "Loser" genes were adopted from bona fide "loser signatures" established in mouse embryos in the study by Lima et al (2021).

### Data processing and clustering with the module score

We analyzed the scRNA-seq data in R (v4.3.1) (URL https://www.R-project.org/) using Seurat (v4.3.0.1) (https://www.rdocumentation.org/packages/Seurat/versions/4.3.0.1) (Hao et al, 2021), beginning with the PercentageFeatureSet function to identify and filter β-cells based on HPAP annotation. A total of 44,559 β-cells were analyzed after being classified as either "Controls" (27,160 cells), "AAB" (8,523 cells), "T1D" (715 cells), or "T2D" (8,161 cells). The NormalizeData function was then used to perform log normalization of the data subsample. The FindVariableFeatures function was used to calculate gene variances and feature variances of standard and clipped values. A variable called "genes.of.interest/loser signature" was created in the module score, and candidate marker genes that represent key determinants of the "loser" status implicated in the UPR (*DDIT3*, *ATF3*, *PPP1R15A*, *RICTOR*) and oxidative stress *NFE2L2* were then assigned to this variable. The ScaleData function was used to center and scale the data matrix. The AddModuleScore function was used to assign module scores to our subsample of either Control, AAB, T1D, or T2D β-cells based on the candidate genes specified in the "genes.of.interest/loser signature." Two matrices were then created: one matrix containing 0 and negative module scores, and the other matrix containing positive module scores. The FindMarkers function was used to find DEGs within each disease state subsample by comparing the positive module scores' matrix with the matrix containing 0 or negative module scores. Post-clustering doublet removal was integral to the HPAP data quality control.

### GeoMx spatial transcriptomics

A geospatial technology platform (GeoMx from NanoString Technologies, Inc.) was used to perform targeted transcriptomic profiling of β-cell populations from three pancreas donors from the

nPOD collection based upon PFKFB3 expression at the Molecular Pathology at City-of-Hope National Center, Duarte, California. We used a ssDNA-labeled probe with photocleavable indexing oligos to bind targets of interest in nPOD pancreatic sections. We also used PFKFB3 and insulin antibodies conjugated with fluorophores as morphology markers to visualize PFKFB3-positive β-cells. ROIs with PFKFB3-positive and PFKFB3-negative β-cells within a single ROI (>100 cells) were segmented and collected separately from each slide from three T2D donors (nPOD #6186, #6255, and #6275). We analyzed 12–16 ROIs from each T2D formalin-fixed pancreatic section (8 PFKFB3-positive and 8 PFKFB3-negative ROIs). ROIs were sequentially exposed to UV light to release the indexing oligos, which were collected and subsequently enumerated via next-generation sequencing. We performed differential profiling of PFKFB3-positive relative to PFKFB3-negative β-cells using GeoMx DSP software for Whole Transcriptome Atlas.

## Gene enrichment analysis

To identify gene expression modules that showed clear, interpretable enrichment for biological functions, we performed a gene enrichment analysis using DAVID (Huang et al, 2009; Sherman et al, 2022) based on Reactome annotations that contained multiple modules with significantly enriched gene ontology profiles (adjusted $P < 0.05$).

To avoid a potential bias resulting from differences between reference cells (β-cells) from all cells, the sets of DEGs (false discovery rate [FDR] < 5%) in Control, AAB, T1D, and T2D β-cells were analyzed with DAVID (Huang et al, 2009; Sherman et al, 2022), using the union of genes observed in all the samples as the reference. Enriched Reactome pathways identified in the individual datasets (FDR <5%) were visualized with Cytoscape (Shannon et al, 2003).

## Cell-specific gene network analysis

We analyzed two independent datasets, β-cell populations from three pancreas donors from the nPOD collection based upon PFKFB3 expression in spatial transcriptomics and scRNA-seq data under four conditions from a β-cell population from the PANC-DB data portal of the HPAP (Kaestner et al, 2019; Montemurro et al, 2019). Weighted Gene Correlation Network Analysis (WGCNA) (Langfelder & Horvath, 2008) was used to construct correlated gene modules. Sample quality control was performed with the WGCNA package function, goodSampleGenes. The soft power threshold was adjusted for each condition to reach scale-free topology. Highly correlated modules among different blocks were then merged to create the final network. Subsequent analysis was focused on the significantly different modules with criteria $P < 0.05$. The top 10 genes for each significant module were highlighted. Single-cell WGCNA was performed using high-dimension (hd)WGCNA (Morabito et al, 2023), with the PCA used for dimensionality reduction, the maximum number of cells shared between two metacells limited to 10, single-cell transform of the expression data, and soft power threshold adjusted to reach scale-free topology (12 for the β-cell subset) as a signed networkType with different values for the individual, that is,

Control, AAB, T1D, and T2D conditions. The top 10 hub genes for each module were highlighted. To reduce the noise of the correlations in the adjacency matrix, soft power thresholds of 12, 16, 20, and 14 were picked for mean, median, and max connectivity to reach the Scale Free Topology Model Fit greater than 0.8 for Control[HPAP], AAB[HPAP], T1D[HPAP], and T2D[HPAP], respectively, of hub genes in the hdWGCNA.

Gene expression analysis was performed using R (v4.3.1) (https://www.R-project.org/). Network visualization was performed with Mathematica (v12; Wolfram Research, Inc.) using GravityEmbedding for the graph layout.

## Mergeomics

We performed weighted Key Driver Analysis (Arneson et al, 2016; Ding et al, 2021) on the subsets of DEGs from geospatial targeted transcriptomic profiling of β-cell populations from three nPOD pancreas donors and from "loser signature" transcriptomic profiling of HPAP scRNA-seq on the University of California, Los Angeles Web Server for Multidimensional Data Integration (http://mergeomics.research.idre.ucla.edu/home.php#). We used the Mergeomics 2.0 Web Server for Multiomics Data Integration (Ding et al, 2021).

## STRING analysis

STRING analysis was performed on the STRING Web portal https://string-db.org using DEGs from the two independent datasets (HPAP and nPOD) (Szklarczyk et al, 2021a, 2021b).

## GGI analysis

Whole-genome sequencing data from the HPAP were downloaded from the NIH dbGAP. Exonic variants were annotated and filtered for genes of interest using bcftools. We used bcftools version 1.11, and as a reference genome, we used the Genome Reference Consortium Human Build 39 patch release 13 (GRCh38.p13 or hg38). All variants for the gene PFKFB3 were filtered out except for the variant of interest, *rs1983890*. To detect all possible pairwise interactions between *rs1983890* and other genes, we used GeneGeneInteR R package and employed the PCA method. For this method, a likelihood-ratio test is conducted to compare two generalized linear models, $M_{inter}$ and $M_0$, where $M_{inter}$ includes an interaction term and is defined as

$$\beta_0 + \sum_{i=1}^{n1} PC_{x1}^i + \sum_{i=1}^{n2} PC_{x2}^j + \sum_{i=1}^{n1} \sum_{i=2}^{n2} PC_{x1}^i PC_{x2}^j$$

and $M_0$ as

$$\beta_0 + \sum_{i=1}^{n1} PC_{x1}^i + \sum_{i=1}^{n2} PC_{x2}^j$$

For both models, $PC_{x1}^i$ refers to the $i$th principal component of all SNPs in gene 1, and $PC_{x2}^j$ to the $j$th principal component of all SNPs in gene 2. The number of principal components retained for each

model, *n1* and *n2*, is found by the percentage of inertia calculated by PCA.

After interaction testing, the resulting *P*-values were adjusted for multiple testing to obtain FDR using the Benjamini–Hochberg procedure.

## Animals

The study received ethical approval ARC-2019-011-AM-004 from the UCLA Animal Research Committee.

The β-cell–specific inducible PFKFB3 knockout or PFKFB3 WT mouse model (*RIP-CreERT:PFKFB3^{fl/fl}*) on *hIAPP^±* background (h-βTG) and HFD (PFKFB3^{βKO} DS) (#D12492; Research Diets Inc.) was previously described (Min et al, 2022) and used as a positive control. At 6–8 wk of age, all mice from the WT or DS group (h-βTG) were assigned to receive the HFD. In addition, mice from the DS group were randomly assigned to a group that received vehicle (DS+vehicle) or the group that received the PFKFB3 inhibitor AZ67 (DS+iPFKFB3) (Tocris Bioscience). For the comparative evaluation of the effect of the PFKFB3 knockdown with the effect of PFKFB3 inhibition, we used AZ67 prepared in 10% absolute ethanol in sunflower oil, matched to the preparation of tamoxifen for the Cre-recombinase induction and PFKFB3 knockout. We randomized the PFKFB3 targeting group into mice (n = 3) that received tamoxifen injection to undergo β-cell–specific PFKFB3 knockout and mice (n = 2) that received AZ67. For the rest of the experiments (two biological replicate experiments; each n = 4–8/group), we used AZ67 prepared in 8% wt/vol 2-hydroxypropyl-beta-cyclodextrin in PBS, pH 7.4, at 28 mg/kg body weight intraperitoneally every day for 7 wk (Fig 7A). The mice had ad libitum access to food and water for the duration of the study. Body weights and fasting blood glucose levels were assessed weekly.

## Intraperitoneal glucose tolerance test (IP-GTT)

An IP-GTT was performed at 4 and 6 wk after the start of the HFD and AZ67 administration. Mice were fasted overnight in a clean cage and with access to water before the analysis. Tail vein blood glucose was collected before and 30, 60, and 120 min after an injection of 20% glucose bolus (2 g/kg of body weight). Fasting blood glucose was measured weekly after overnight fasting for 15 h. The blood glucose levels were measured in tail-drawn blood through the use of a FreeStyle blood glucometer (Abbott Diabetes Care Inc.).

## Pancreas perfusion and isolation

Mice were euthanized by a brief isoflurane exposure before cervical dislocation. A medial cut was made to open the abdomen and chest cavities. A cut of the right ventricle was followed by a poke of the left ventricle with a needle to inject 10 ml of cold PBS slowly for perfusion of the pancreas. After perfusion, the pancreas was placed in cold PBS and separated from other tissues, including the surrounding fat. The pancreas was then weighed before and after the excess PBS had been absorbed into the lint-free tissue.

## Histological assessments

After the excision of smaller pieces, the pancreas was fixed overnight at 4°C in 4% PFA (19202; Electron Microscopy Sciences). The pancreas was paraffin-embedded and sectioned into 4-μm-thick slices by the Translational Pathology Core Laboratory at UCLA.

Immunofluorescence analysis was performed using Openlab 5.5.0 software on the Leica DM6000 B research microscope. The following antibodies were used: guinea pig anti-insulin (ab195956, 1:400; Abcam); mouse anti-glucagon (G2654, 1:1,000; Sigma-Aldrich); mouse anti-c-Myc (9E10, sc-40, 1:100; Santa Cruz Biotechnology Inc.); and mouse anti-HIF1α (NB100-105, 1:50; Novus Biologicals), anti-MAIP1 (D-12, 1:50; Santa Cruz Biotechnology), and anti-AGAP1 (MBS5312069, 1:50; MyBioSource Inc.).The following secondary antibodies were used: F(ab')2 conjugates with fluorescein iso-thiocyanate donkey anti-guinea pig immunoglobulin G (IgG) (heavy and light, H + L) (706-096-148, 1:200 for intrinsic factor [IF]; Jackson ImmunoResearch) and F(ab')2 conjugates with Cy3 donkey anti-mouse IgG (H + L) (711-165-151, 1:200 for IF; Jackson Immuno-Research). Vectashield containing 4′,6-diamidino-2-phenylindole (DAPI) (H1200; Vector Laboratories) was used to mount the slides. Imaging and data analysis were performed by two observers in a blinded fashion for each section of the experimental mouse genotype. The morphometric data for PFKFB3–, MAIP1– and AGAP1 positive cells in experimental mice and MAIP1 positive β-cells in human pancreata from nPOD is presented in Table S14.

## Statistical analysis

Data are presented as errors of the means (standard error, SEM) for the number of mice indicated in the figure legends. Mean data were compared between groups by one-way analysis of variance (ANOVA) followed by Tukey's or Dunnett's post hoc test for multiple comparisons. *P*-values less than 0.05 were considered significant.

# Data Availability

All data are available in the main text or the supplementary materials. The custom R script code with the module score used for this study is available on the GitHub platform (https://github.com/KavitRaval/Loser-Vs-Winner-beta-cells_Rscript/commit/4d7ace19ed6bf89f49f55b0b2bb386c566623137).

# Supplementary Information

# Acknowledgements

This study used data acquired from the Human Pancreas Analysis Program (HPAP-RRID:SCR_016202) database (https://hpap.pmacs.upenn.edu), a Human Islet Research Network (RRID:SCR_014393) consortium (UC4-DK-112217, U01-DK-123594, UC4-DK-112232, and U01-DK-123716). The HPAP study

datasets were accessed with appropriate approval through the dbGaP online resource (https://dbgap.ncbi.nlm.nih.gov/aa/wga.cgi?page=login) (phs002465.v1.p1). This work was supported by the Larry Hillblom Foundation (Start-up Grant #2017-D-002-SUP) (to S Tudzarova), Hirshberg Foundation Seed Grant HF-2023-024 (to S Tudzarova), and Sponsored Research Agreement 2021-0206 between UCLA and Metanoia Bio Inc. (to S Tudzarova and K Raval).

## Author Contributions

K Raval: data curation, formal analysis, validation, visualization, and writing—review and editing.
N Jamshidi: data curation, software, formal analysis, validation, methodology, and writing—review and editing.
B Seyran: formal analysis and methodology.
L Salwinski: data curation, visualization, and methodology.
R Pillai: data curation, formal analysis, and methodology.
L Yang: validation, visualization, and methodology.
F Ma: software, formal analysis, validation, visualization, and methodology.
M Pellegrini: software, formal analysis, validation, and methodology.
J Shin: software, visualization, and methodology.
X Yang: software, formal analysis, validation, and methodology.
S Tudzarova: conceptualization, resources, formal analysis, supervision, funding acquisition, validation, investigation, visualization, project administration, and writing—original draft, review, and editing.

## Conflict of Interest Statement

The authors declare that they have no conflict of interest.

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
