## [Reviewer comments · Life Science Alliance]

Life Science Alliance

Dysfunctional β -cell longevity in diabetes relies on high energy conservation and positive epistasis

Slavica Tudzarova, Kavit Raval, Neema Jamshidi, Berfin Seyran, Lukasz Salwinski, Raju Pillai, Lixin Yang, Feiyang Ma, Matteo Pellegrini, Juliana Shin, and Xia Yang

DOI: <https://doi.org/10.26508/lsa.202402743>

Corresponding author(s): Slavica Tudzarova, University of California, Los Angeles

Review Timeline:

Submission Date:	2024-03-27
Editorial Decision:	2024-06-04
Revision Received:	2024-09-04
Editorial Decision:	2024-09-05
Revision Received:	2024-09-11
Accepted:	2024-09-12

Transaction Report:

June 4, 2024

Re: Life Science Alliance manuscript #LSA-2024-02743-T

Dr. Slavica Tudzarova
University of California Los Angeles
Medicine
900 Weyburn Place
CHS 33-165
Los Angeles, California 90024

Dear Dr. Tudzarova,

Thank you for submitting your manuscript entitled "Dysfunctional β -cell longevity in diabetes relies on high energy conservation and positive epistasis" to Life Science Alliance. The manuscript was assessed by expert reviewers, whose comments are appended to this letter. We invite you to submit a revised manuscript addressing the Reviewer comments.

Thank you for this interesting contribution to Life Science Alliance. We are looking forward to receiving your revised manuscript.

Sincerely,

B. MANUSCRIPT ORGANIZATION AND FORMATTING:

Reviewer #1 (Comments to the Authors (Required)):

This article discusses the phenomenon of dysfunctional β -cell longevity in diabetes and its reliance on high energy conservation and positive epistasis. The study explores the concept of cell fitness competition (CFC) and its role in maintaining tissue function by selectively clearing unhealthy β -cells. Through the analysis of human β -cells in diabetes, the researchers identify specific molecular characteristics of dysfunctional "loser" β -cells, including downregulation of ribosomal biogenesis and genes related to mitochondrial respiration. They also find an interaction between the PFKFB3 gene and an anti-apoptotic gene, suggesting a mechanism for the prolonged survival of dysfunctional β -cells. Inhibition of PFKFB3 leads to the clearance of dysfunctional β -cells and improves glucose tolerance in a mouse model of diabetes.

The article provides a clear overview of the research conducted on dysfunctional β -cell longevity in diabetes. The introduction effectively sets the context and explains the rationale behind the study. The content is organized logically and follows a structured approach. I would recommend it for publication.

I have a few concerns:

- 1- Notable, the article highlights the potential role of cell fitness competition (CFC) and positive epistasis in the longevity of dysfunctional β -cells in type-2 diabetes (T2D). This finding has important implications for understanding the pathology of T2D and potentially developing therapeutic interventions. However, the article does not discuss the broader implications of these findings or suggest specific applications in clinical practice.
- 2- More, the article mentions the downregulation of ribosomal biogenesis and genes encoding mitochondrial respiration in "loser" β -cells but does not elaborate on the potential mechanisms or functional consequences of these changes. Providing more detailed discussion and interpretation of the results would enrich the article.
- 3- The article is generally well-written and uses appropriate scientific terminology. However, there are a few instances where the language could be further simplified or clarified to improve readability for a broader audience.

Reviewer #2 (Comments to the Authors (Required)):

Manuscript# LSA-2024-02743-T

This is an interesting study by Raval and colleagues proposing that PFKFB3 is a biomarker of dysfunctional β -cells. The authors provide evidence in support of the notion that the accumulation of PFKFB3+ β -cells might be at the very heart of a type 2 diabetes metabolic state.

Importantly, through a variety of sophisticated transcriptomic analytical tools, they define key transcriptional drivers of so-called "loser" β -cells, and provide some evidence that the systemic inhibition of PFKFB3 by the small molecule AZ67 improves metabolic performance of β -cells in a model of type 2 diabetes. Collectively, this is a study that has important translational for type 2 diabetes.

There are however a number of issues that need to be addressed:

Based on the fact that most of the study focuses on analytical tools, some of the conclusions are essentially based on the analysis interpretation of bioinformatic data published by others, and not on any actual validation of the suggested mechanism of epistasis via MAIP1 and AGAP1. It would be important to show that MAIP1 and AGAP1 are regulating this process in the animal model that the authors have used for some of the data shown in Figures 5, 6, and EV1.

The manuscript is populated by a tsunami of acronyms that are not defined making the reading very difficult.

Legend to Figure 2 does not identify panel A.

In Figure 5, the labeling of panel E is confusing. The images are presented as "representative", but there is no indication or

mention of the number of experimental determinations, the number of pancreatic sections stained and analyzed. The quantitative data presented in panel B, C and D suggest that the authors have only stained 3 to 5 sections, or analyzed 3 to 5 images to quantify the frequency of bihormonal cells, c-Myc+ and HIF1a+ cells. These are important morphometric data that need to be added.

The discussion is too long, and could be reduced by at least a third.

Manuscript#LSA-2024-02743-T

Raval et al. "Dysfunctional β -cell longevity in diabetes relies on high energy conservation and positive epistasis"

Reviewer#1:

We thank Reviewer #1 for his supportive and constructive review and comments as well as for seeking to address important implications of our paper. In relation to Reviewer's 1 comments please find here below:

Color code:

Black - reviewer's comments (R).

Blue - author's response (A).

Orange - citations from the manuscript referring to the amended manuscript section.

All the revision amendments in the manuscript text are highlighted in grey.

R1- Notable, the article highlights the potential role of cell fitness competition (CFC) and positive epistasis in the longevity of dysfunctional β -cells in type-2 diabetes (T2D). This finding has important implications for understanding the pathology of T2D and potentially developing therapeutic interventions. However, the article does not discuss the broader implications of these findings or suggest specific applications in clinical practice.

A1-We completely agree with the reviewer, and we now devoted a paragraph in the Discussion explaining the broader implications of our findings and suggesting specific applications in the clinical practice (page 21, first paragraph). Please see the section copied from the Manuscript below quoted:

"Selective clearance of dysfunctional "loser" β -cells agnostic of their origin (e.g. senescent cells) by reactivation of physiologically-competent CFC is an original approach. This study predicts that T2D patients without PFKFB3 polymorphism should have a better clinical outcome, what remains to be addressed in the future. Pharmacological targeting of PFKFB3 to

abort survival of dysfunctional "loser" β -cells holds a promise to change and prevent deterioration of early T2D trajectory given that at this stage both the prevalence of "loser" β -cells and "loser" β -cell survival selectively depend on PFKFB3. The therapeutic ramification of CFC in T2D is emerging with a potential to modify this disease by enrichment with functional β -cells particularly at early-onset where β -cell mass is still conserved. This approach when therapeutically developed may find application in other age-related diseases (e.g. neurodegeneration) and aging itself, given the shared feature of dysfunctional cell accumulation across many tissues in addition to pancreas."

R2- More, the article mentions the downregulation of ribosomal biogenesis and genes encoding mitochondrial respiration in "loser" β -cells but does not elaborate on the potential mechanisms or

functional consequences of these changes. Providing more detailed discussion and interpretation of the results would enrich the article.

A2-Thank you so much for this great suggestion. We completely agree with the reviewer, and we now devoted a paragraph in the Discussion explaining the potential mechanisms and functional consequences of these changes (page 18, paragraph 2, and page 19, paragraph 1). We agree that this addition greatly enriched our article.

“The profound downregulation of RiBi and genes of mitochondrial respiration in “loser” β -cells indicates that conserving energy by shutting down the energy-expensive process of making proteins is a priority of long-lived dysfunctional β -cells. This could be for example, critical for preserving proteosynthetic machinery and orphaned ribosomal proteins, considered a prerequisite for an intact restart of RiBi once the stress is resolved. (Buchan & Parker, 2009; Wheeler *et al*, 2016). Why is the energy conservation necessary in “loser” β -cells? Previously it was demonstrated that downregulation of RiBi or metabolism reduction rescued lethal *Drosophila* mutants during development due to redundance of repressor activities under those conditions (Cassidy *et al*, 2019). Therefore, strained transcriptional regulation and genomic stability under stress can be protected from deleterious consequences under energy conservation. The benefit of energy conservation known as hypometabolism on organismal level under environmental constraints was demonstrated with torpor or hibernation (Giroud *et al*, 2020) and is currently considered as a radiation injury-preventative approach for the human space flights (Ghosh *et al*, 2017).”

R3- The article is generally well-written and uses appropriate scientific terminology. However, there are a few instances where the language could be further simplified or clarified to improve readability for a broader audience.

A3-We have now checked all the acronyms, we explained them and tried to simplify them (this is present in track changes across the whole Manuscript).

Reviewer #2

This is an interesting study by Raval and colleagues proposing that PFKFB3 is a biomarker of dysfunctional β -cells. The authors provide evidence in support of the notion that the accumulation of PFKFB3+ β -cells might be at the very heart of a type 2 diabetes metabolic state. Importantly, through a variety of sophisticated transcriptomic analytical tools, they define key transcriptional drivers of so-called "loser" β -cells and provide some evidence that the systemic inhibition of PFKFB3 by the small molecule AZ67 improves metabolic performance of β -cells in a model of type 2 diabetes. Collectively, this is a study that has important translational for type 2 diabetes.

We thank Reviewer #2 for the sharp analysis and great suggestions which we addressed as follows:

R1-Based on the fact that most of the study focuses on analytical tools, some of the conclusions are essentially based on the analysis interpretation of bioinformatic data published by others, and not on any actual validation of the suggested mechanism of epistasis via MAIP1 and AGAP1. It would be important to show that MAIP1 and AGAP1 are regulating this process in the animal model that the authors have used for some of the data shown in Figures 5, 6, and EV1.

A1-We thank the reviewer for this great suggestion that strengthened the foundation of this paper. Here we performed additional analyses in h- β TG mouse model of T2D.

- a) **MAIP1.** MAIP1 is a novel anti-apoptotic protein that can limit the Ca^{2+} influx in the mitochondria under stress which prevents the opening of the mitochondrial permeability transition pore. We have performed immunostaining of MAIP1 in β -hTG model of T2D. We found that under diabetogenic stress there is an islet specific increase in MAIP1 expression exclusively in β -cells with reduced insulin immunopositivity and higher PFKFB3 expression. A threshold in expression levels of MAIP1 and PFKFB3 is necessary for epistasis and their co-localization indicates that the two act in tandem when protecting dysfunctional β -cells from clearance by CFC. We confirmed their co-localization using serial sections and qualitative and semiquantitative comparison of the same islets immunostained either for MAIP1 and PFKFB3 in β -hTG model and human pancreata from the nPOD collection of T2D. Ca^{2+} toxicity is a converging point of diabetic stressors and the sharp difference between human non-diabetic and T2D islets positions MAIP1 as a novel diabetes β -cell marker. When it comes to demonstrate epistasis, we need to analyze T2D diabetic patients with and without PFKFB3 polymorphism. Our study predicts that T2D patients without PFKFB3 polymorphism should have a better clinical outcome given the lesser probability to accumulate dysfunctional β -cells, what remains to be addressed in the future.
- b) Also, we plan loss-of-function studies on MAIP1 which is not in the scope of this paper, and which will determine whether MAIP1 knockout leads to the same outcome as PFKFB3 knockout or inhibition. If so, MAIP1 may have an equivalent importance to PFKFB3 as a novel target that can unlock CFC in T2D.

Please see the inserted Figure 6 that demonstrates our studies of MAIP1 in the β -hTG mouse model and in pancreata from human T2D donors.

Fig. 6 A Representative immunofluorescence images of islets from mice as described under Fig. 5, either double immunostained for insulin- and MAIP1

B or immunostained for insulin and PFKFB3; (all markers in red), insulin (green) and nuclei (blue) showing eradication of MAIP1 and PFKFB3 positive β -cells in WT+HFD and DS+iPFKFB3 mice. Scale bar: 50 μ m.

C Insets from identical serial sections showing MAIP1 and high PFKFB3 immunostaining overlap. White arrows indicate β -cells with PFKFB3 and MAIP1 high expression overlap. Scale bar: 20 μ m

D Representative immunofluorescence images of islets from human non-diabetic (ND) and T2D donors immunostained for insulin- and MAIP1. Scale bar: 50 μ m.

E Quantitative representation of MAIP1 (circles) and PFKFB3 (triangles) positive β -cells (% of all β -cells) in mice groups as described under Fig. 5. n=3, data is represented as mean \pm SEM, p<0.05.

F Quantitative representation of MAIP1 positive β -cells (% of all β -cells) in human non-diabetic (ND) and T2D donors. n=3, data is represented as mean \pm SEM, p<0.05.

G Quantitative representation of AGAP1 positive non-insulin-islet cells (% of all non-insulin-islet cells) in mice groups as described in Fig. 5. n=3, data is represented as mean \pm SEM, p<0.05.

Figure S14

Supplementary Fig. S14 Identical islet sections from DS+vehicle show overlap between MAIP1 and PFKFB3
Identical islets from serial sections were immunostained for MAIP1 and PFKFB3 and the overlap was semi-quantitatively confirmed.

A Representative immunofluorescence images of islets from DS+vehicle mice, double immunostained for insulin- and MAIP1. Scale bar: 20 μ m

B Representative immunofluorescence images of islets from DS+vehicle mice double immunostained for insulin- and PFKFB3. Scale bar: 20 μ m.

c) **AGAP1**. AGAP1 is an important gene in endosomal protein trafficking and cytoskeleton organization. However Human Protein Atlas indicates that AGAP1 is not expressed in β -cells. Our immunostaining confirms that AGAP1 was expressed in non-insulin positive islet cells and interestingly, AGAP1 was not affected neither by diabetogenic stress nor after inhibition of PFKFB3. Interestingly, AGAP1 antibody strongly immunostained blood vessels and the AGAP1 gene-gene interaction with PFKFB3 may exert effect on blood vessels where PFKFB3 plays a critical role in vessel sprouting and angiogenesis. This interaction becomes very interesting in context of diabetic microcapillary degeneration and microvasculopathies. As such, it is outside of the scope of this paper.

Figure S15

Supplementary Fig. S15 AGAP1 expression in non-insulin positive islet cells

A Representative immunofluorescence images from split fluorescent channels from mouse islets (WT+HFD, DS+vehicle and DS+iPFKFB3) double immunostained for insulin- and AGAP1 showing comparable frequency of AGAP1 in non-insulin islet cells. Scale bar: 50 μ m.

B Enlarged merged images from mouse islets (WT+HFD, DS+vehicle and DS+iPFKFB3 mice) double immunostained for insulin- and AGAP1 showing comparable frequency of AGAP1 in non-insulin islet cells. Scale bar: 50 μ m.

R2-The manuscript is populated by a tsunami of acronyms that are not defined making the reading very difficult.

A2-We agree with the reviewer, and we now carefully went through all the acronyms and defined them.

R3-Legend to Figure 2 does not identify panel A.

A3-We apologize for this omission. We have now corrected and identified panel A. See below the citation in the manuscript:

“A An experimental evidence (>90% confidence level) based protein-protein interaction framework of DEGs with a cluster of ribosomal biosynthesis and absence of the cluster for mitochondrial respiration in AAB^{HPAP}.”

R4-In Figure 5, the labeling of panel E is confusing. The images are presented as "representative", but there is no indication or mention of the number of experimental determinations, the number of pancreatic sections stained and analyzed. The quantitative data presented in panel B, C and D suggest that the authors have only stained 3 to 5 sections, or analyzed 3 to 5 images to quantify the frequency of bihormonal cells, c-Myc+ and HIF1 α + cells. These are important morphometric data that need to be added.

A4-We fully agree with the reviewer, and we now explained the statistics from the morphometric analysis. Please see below the citation from the clarification now included in the manuscript (from the Figure Legends):

“The sectioning was performed across the whole pancreas. For quantification of the markers’ frequency the whole pancreas section per treatment group was used. We have counted all the pancreatic islets from each section, i.e 170 to 200 islets per mouse group (n=3).

R5-The discussion is too long and could be reduced by at least a third.

A5-We fully agree with the reviewer, and we now reduced the discussion as recommended by the reviewer. We, however, included the clinical implications of our study in the discussion as recommended by the reviewers.

September 5, 2024

RE: Life Science Alliance Manuscript #LSA-2024-02743-TR

Dr. Slavica Tudzarova
University of California, Los Angeles
Medicine
10833 Le Conte Avenue
CHS 33-165
Los Angeles, California 90095

Dear Dr. Tudzarova,

Thank you for submitting your revised manuscript entitled "Dysfunctional β -cell longevity in diabetes relies on high energy conservation and positive epistasis". We would be happy to publish your paper in Life Science Alliance pending final revisions necessary to meet our formatting guidelines.

- please be sure that the authorship listing and order is correct
- please add the Twitter handle of your host institute/organization as well as your own or/and one of the authors in our system
- please add your supplemental figure legends and your table legends to the main manuscript text, directly after the legends for the main figures
- we encourage you to introduce your figure panels in the figure legends in alphabetical order
- on page 10, there is a figure callout with figure panels, but the figure number is missing. Please correct.
- please include accession information for the RScript code in the Data Availability statement

Figure Check:

- please add scale bars to the top images of Figure S1

LSA now encourages authors to provide a 30-60 second video where the study is briefly explained. We will use these videos on social media to promote the published paper and the presenting author (for examples, see <https://docs.google.com/document/d/1-UWCfbE4pGcDdcgzcmiuJl2XMBJnxKYeqRvLLrLS08s/edit?usp=sharing>). Corresponding or first-authors are welcome to submit the video. Please submit only one video per manuscript. The video can be emailed to contact@life-science-alliance.org

A. FINAL FILES:

B. MANUSCRIPT ORGANIZATION AND FORMATTING:

Sincerely,

September 12, 2024

RE: Life Science Alliance Manuscript #LSA-2024-02743-TRR

Dr. Slavica Tudzarova
University of California, Los Angeles
Medicine
10833 Le Conte Avenue
CHS 33-165
Los Angeles, California 90095

Dear Dr. Tudzarova,

Thank you for submitting your Research Article entitled "Dysfunctional β -cell longevity in diabetes relies on high energy conservation and positive epistasis". It is a pleasure to let you know that your manuscript is now accepted for publication in Life Science Alliance. Congratulations on this interesting work.

DISTRIBUTION OF MATERIALS:

Again, congratulations on a very nice paper. I hope you found the review process to be constructive and are pleased with how the manuscript was handled editorially. We look forward to future exciting submissions from your lab.

Sincerely,
